# On the application and grid-size sensitivity of the urban dispersion model CAIRDIO v2.0 under real city weather conditions

Michael Weger[1], Holger Baars[1], Henriette Gebauer[1], Maik Merkel[1], Alfred Wiedensohler[1], and Bernd Heinold[1]

[1]Leibniz Institute for Tropospheric Research, Leipzig, Germany

**Correspondence:** Michael Weger (weger@tropos.de)

**Abstract.** There is a gap between the need for city-wide air-quality simulations considering the intra-urban variability and mircoscale dispersion features and the computational capacities that conventional urban microscale models require. This gap can be bridged by targeting model applications on the gray zone situated between the mesoscale and large-eddy scale. The urban dispersion model CAIRDIO is a new contribution to the class of computational-fluid dynamics models operating in this scale range. It uses a diffuse-obstacle boundary method to represent buildings as physical obstacles at gray-zone resolutions in the order of tens of meters. The main objective of this approach is to find an acceptable compromise between computationally inexpensive grid sizes for spatially comprehensive applications and the required accuracy in the description of building and boundary-layer effects. In this paper, CAIRDIO is applied on the simulation of black carbon and particulate matter dispersion for an entire mid-size city using an uniform horizontal grid spacing of $40\,\mathrm{m}$. For model evaluation, measurements from 5 operational air monitoring stations representative for the urban background and high-traffic roads are used. The comparison also includes the mesoscale host simulation, which provides the boundary conditions. The measurements show a dominant influence of the mixing layer evolution at background sites, and therefore both the mesoscale and LES simulation results are in good agreement with the observed air pollution levels. In contrast, at the high-traffic sites the proximity to emissions and the interactions with the building environment lead to a significantly amplified diurnal variability in pollutant concentrations. These urban road conditions can only be reasonably well represented by CAIRDIO while the meosocale simulation indiscriminately reproduces a typical urban-background profile, resulting in a large positive model bias. Remaining model discrepancies are further addressed by a grid-spacing sensitivity study using offline-nested refined domains. The results show that modeled peak concentrations within street canyons can be further improved by decreasing the horizontal grid spacing down to $10\,\mathrm{m}$, but not beyond. Obviously, the default grid spacing of $40\,\mathrm{m}$ is too coarse to represent the specific environment within narrow street canyons. The accuracy gains from the grid refinements are still only modest compared to the remaining model error, which to a large extend can be attributed to uncertainties in the emissions. Finally, the study shows that the proposed gray-scale modeling is a promising downscaling approach for urban air-quality applications. The results, however, also show that aspects other than the actual resolution of flow patterns and numerical effects can determine the simulations at the urban microscale.

# 1 Introduction

Air pollution from particulate matter (PM) is a major risk factor to population health and is estimated to contribute to at least 8 million premature deaths annually (Burnett et al., 2018; Vohra et al., 2021). For Europe, similar figures estimate the annual mortality excess by up to 900,000 (Tarín-Carrasco et al., 2021), thus health-adverse air quality conditions remain an issue also in well-developed countries. Despite continued efforts in emission reductions and an associated decline in $PM_{2.5}$ concentrations over the past decades (Ortiz and Guerreiro, 2020), the associated mortality risk has only slowly responded to it, and it may increase again in the future due to projected changes in demographics and climate in an adverse way to increase population vulnerability (Sicard et al., 2021). The burden on human health from air pollution is especially relevant for urban areas, not only because more than half of the global population resides there. Urban emissions from traffic and industrial locations locally contribute to a large extend to the primary particulate matter (PPM) fraction, which is most hazardous to human health (Park et al., 2018). Important health-relevant constituents are black-carbon (BC), organic carbon (OC) and road dust. PPM typically exhibits high concentrations close to sources, rapidly decreasing with increased distance, and thus also largely determines the intra-urban variability of PM (Wu et al., 2015). On the other hand, secondary particulate matter (SPM) is formed within the atmosphere from precursor compounds, is generally considered less toxic (Park et al., 2018), and transport is more relevant at the regional scale (Ying et al., 2021), thus taking effect as a near-constant offset in PM concentrations at the intra-urban scale. Aside from the mere location of sources, the effect of the urban canopy on pollutant dispersion is a crucial factor in the characteristic of the intra-urban variability of air pollution (Brown et al., 2015). Buildings affect pollution dispersion across multiple scales through mechanical and thermal interactions with the air flow, which are often too complex to be described in a general way (Roth, 2000). As a consequence, a large pool of literature exists on this topic. Studies investigating such effects are either based on experimental observations, physical models, numerical simulations or combined approaches of varying complexity. To mention few examples, studies have been carried out for isolated buildings (Higson et al., 1996; Foroutan et al., 2018; Jiang and Yoshie, 2020), arrays of mounted obstacles (Coceal et al., 2014; Fuka et al., 2018; Goulart et al., 2019), and urban canopies close to reality (Auguste et al., 2020; Hertwig et al., 2021). While for single buildings, the number of variables reduces to the shape and orientation of the obstacle relative to the approaching flow, the relative position of obstacles to each other (aligned vs. staggered) is at least as important for building arrays. In the aligned case, buildings may contribute to horizontal dispersion by channelling the mean flow through wind-parallel street-canyons, which can be further enhanced by isolated tall buildings (Fuka et al., 2018). On the other hand, transversal and vertical dispersion is more pronounced in the staggered case. For vertical dispersion, both the contributions by the mean flow from the diverging streamlines in front of the upstream faces and turbulent dispersion in the building wakes are of similar importance (Goulart et al., 2019). Buildings do not only enhance dispersion but can also trap pollution within horizontal re-circulation zones, which then act as secondary sources (Coceal et al., 2014). Furthermore, while vertical transport in proximity to sources leads to an efficient detrainment of air pollution from near the surface, it may be re-introduced further downstream through down-washing processes, especially in the vicinity of tall buildings (Goulart et al., 2018). Trees within street canyons act as additional momentum sinks, which in turn have been shown to impair ventilation by slowing-down or disturbing the street-canyon vortex (Li and Wang, 2018).

The importance of an explicit model-representation of urban trees compared to a simpler surface-roughness parameterization or the absence of any tree effects was investigated by Salim et al. (2015). On a larger scale, the extraction of energy from the mean flow by buildings causes the roughness sublayer, within air pollutants are efficiently mixed, to grow in thickness. This effect is especially pronounced for a heterogeneous building-height distribution (Hertwig et al., 2021; Makedonas et al., 2021). Aside from the purely mechanical effects, the buoyancy effects from heated building walls and enclosed ground surfaces also have an important effect on the air-exchange rate across the roof level. Various studies have investigated such effects either in the framework of field campaigns (Louka et al., 2002) or wind-tunnel experiments (Allegrini et al., 2013; Marucci and Carpentieri, 2019). For a uniform heating of urban surfaces, the turbulent air-exchange rate across the roof level is significantly enhanced by buoyancy. These studies also emphasize the effects of differential heating of oppositely orientated building walls on either the intensification or disturbance or splitting of the street canyon vortex, depending on the approach-flow direction. Temperature differences may also occur between roof and ground level and can have similar effects (Park et al., 2016). On a larger scale, anthropogenic heating, radiation trapping and heat storage within building walls increases the magnitude of the urban-heat island effect (Kotthaus and Grimmond, 2014). The positive surface-sensible-heat flux destabilizes the urban planetary boundary layer (PBL), which in combination with wind shear from the mechanical building effects leads to an increase in the PBL top height (hereafter referred to as mixed-layer height, MLH) (Roth, 2000).

Exposure studies depend on modeling of the various aspects in addition to point monitoring observations and, recently, also measurement networks. However, there remains a gap between the need for an accurate estimation of pollutant concentrations at various characteristically unique exposure sites across the urban scale and the limited applicability of consulted model data for this purpose. Mesoscale chemistry-transport models (CTMs) are used to simulate emission, regional transport, deposition and chemistry of a multitude of pollutant species (Wolke et al., 2012; Cames and Eckard, 2013). Therein, the effects of an urban canopy can be included by urban-canopy parameterizations (Martilli et al., 2002; Schubert et al., 2012). Nevertheless, the spatial resolution applied with such models is generally too coarse for a representation of the true magnitude of the intra-urban variability. Also, building effects at the microscale, like pollution trapping or horizontal channeling, cannot be considered in such a parameterized form. As a consequence, urban air pollution fields modeled with CTM's are typically representative to the urban background (Korhonen et al., 2019). Dealing with the necessity to more accurately represent the urban variability, but at the same time to avoid the prohibitively large computational costs of microscale model applications at the meter scale, we presented the urban dispersion model CAIRDIO with diffuse-obstacle boundaries in Weger et al. (2021). Therein, we recognized the model scale gap between the mesoscale and the urban microscale, and the advantage of gray-zone horizontal resolutions between these two scales from a computational perspective. Model comparison with an idealized wind-tunnel experiment, also included in Weger et al. (2021), showed that relevant aforementioned mechanical building effects can be represented to a satisfactory degree for valid dispersion simulations performed at horizontal gray-zone resolutions ($\Delta h = 40\,\mathrm{m}$). Most importantly, buildings influenced dispersion with the mean flow in a correct way for most of the time, even when obstacles are described as diffuse features similar to a porous medium representation. Turbulent fluxes from buildings, on the other hand, are predominantly parameterized at such comparatively coarse resolutions, which makes model results sensitive to the prescribed mixing length.

In this follow-up paper we shift the focus from idealized experiments to a more application-oriented use of the model for a real city and true atmospheric conditions, for which the mid-sized city of Leipzig in eastern Germany is selected as a showcase.

This allows us to include further processes in the model, which are paramount for realistic dispersion simulations within a real urban canopy and realistic meteorology. For example, the stratification of the PBL does not necessarily have to be neutral and can be further modified locally in the model by a parameterized surface-heat flux from ground and building surfaces. Inflow conditions are in general not only turbulent but also transient, in order to account for an accurate evolution of the larger-scale meteorology. The complexity of the simulation is further increased by using a comprehensive emission inventory that includes

all relevant sectors, which are modulated in time to account for diurnal and weekly changes in activity. While this study aims not at analyzing individual processes in depth, its main objectives are to demonstrate the feasibility and practicability of the approach as a downscaling tool for a more accurate representation of the intra-urban air-pollution variability. Therefore, apart from static inputs, the model solely relies on the output fields of a host simulation conducted at the lower end of the mesoscale, for which the CTM COSMO-MUSCAT (Wolke et al., 2012) is used. For validation, we compare modeled $PM_{10}$

and/or BC concentrations with measurements at 5 different operational air-monitoring sites in Leipzig for a total period of two consecutive days in spring 2020. To further estimate the sensitivity to the horizontal grid spacing, locally-nested sensitivity runs are performed, for which the horizontal grid spacing is decreased from a default $\Delta h = 40\,\mathrm{m}$ in steps down to $\Delta h = 5\,\mathrm{m}$, enabling conventional building-resolved simulations.

The paper is structured as follows: Section 2 describes the methodology, in which all the general and technical aspects of the

110 simulations and measurements are described. This also includes a detailed description of the mesoscale coupling. Section 3 includes the presentation and discussion of model results, which is subdivided into a part describing the modeled PBL evolution, a model evaluation with concentration measurements, including a comparison with results from the CTM COSMO-MUSCAT, and the grid-size sensitivity study. Section 4 summarizes the main findings of the study and highlights the advantages but also limitations of the demonstrated approach and the study itself.

**2 Methodology**

### 2.1 Study time period

The model case study spans 2 consecutive days from 1 March 2020, 00:00 UTC to 3 March 2020, 00:00 UTC, to address the main objectives of this study. Yet, for a more significant model evaluation with observational data, a substantially longer simulation period needs to be simulated. While principally our approach is computationally much cheaper compared to a well-

120 resolved urban microscale simulation, a compromise still had to be found, and we decided to invest computation resources in a spatially more comprehensive simulation that fits better the aspect of a model case study. The specific simulation period was selected based on suitable properties for an investigation of the intra-urban air-pollution variability. Firstly, quality-assured observational data from all operational air-monitoring sites in Leipzig are available during this period. Secondly, significant impacts of the world-wide spreading Covid-19 pandemic had still not reached the German public by early March 2020, as

data from the Google mobility report show a significant decline starting after 10 March (Google-LLC, 2020; Forster et al.,

2020). This provided confidence that the traffic emissions had not to be adjusted for a reduced mobility, which is a potential source of additional uncertainty. Thirdly and most importantly, the meteorological conditions were suitable to focus on local air pollution and PBL processes affecting it. The large-scale synoptic pattern during the simulation period from 1 to 3 March 2020 was dominated by a large and deep low-pressure system situated over northwestern Europe, from which two troughs protruded southward and eventually moved across Germany (see Figure 1a-c). The associated unsettled weather conditions in Leipzig resulted in diverse PBL characteristics and effects on local air quality, which are interesting to study. Moreover the influence of low pressure favored low background near-surface $PM_{10}$ concentrations over most of Germany, as suggested by results from an air-quality simulation for Europe depicted in Fig. 1d-f (cf. Section 2.3 for a detailed description of the mesoscale simulations). According to this, highest $PM_{10}$ concentrations apart from the well-known air-polluted regions, like the Po Valley, occurred over the eastern half of Europe. There were also periods before and after the actual simulation period, when the Siberian high pressure system extended westward and brought a polluted continental air mass to central Europe (not shown). During such periods with elevated background concentrations, the intra-urban air-pollution variability was quite insignificant and not worth to study.

## 2.2   Observations

### 2.2.1   Near-surface in-situ observations of air quality

A set of in-situ observations is used to evaluate the modeled air pollutant concentrations. The south-eastern German state Saxony operates 26 air-quality monitoring sites. Three of them are suitable for our model evaluation, as they are located within the city margins of Leipzig and provide $PM_{10}$ concentration measurements. One of the three stations considered is also co-operated by the TROPOS and provides BC retrievals too. Three stations are air quality stations operated by the Saxon State office while two additional stations belong to TROPOS. Three out of the five station measure also equivalent black carbon (eBC) mass concentration, which are also relevant to our evaluation. Tab. 1 lists some basic information of the aforementioned measurement sites, while Fig. 2 shows their locations on horizontal maps of the simulation domains to provide a qualitative overview of the characteristic environment, like the distribution of buildings, parks and major roads highlighted by $PM_{10}$ line emissions. Finally, Fig. 3 gives a detailed mapping of the surrounding building environment in spherical coordinates at each exact measurement position, as well as the corresponding sky-view factors $f_{sky}$. Based on these information, a brief introduction of each measurement site is given in the following. Station Leipzig Lindenau (LL) measures $PM_{10}$ and is located in the western city district Leipzig Lindenau within a closed street canyon running in west-northwest east-southeast direction. The street canyon has average dimensions (height × width) of roughly $20\,\mathrm{m} \times 25\,\mathrm{m}$, which results in a sky-view factor of 0.34. The horizontal position of the measurement site is near the northern side of the canyon at a distance of only $5\,\mathrm{m}$ to a traffic-busy road, with an average daily traffic count (ADTC) of 20400 vehicles. This, in combination with an inlet height of $1.7\,\mathrm{m}$ leads to a high exposure of this station to the exhaust gases from nearby traffic, which clearly classifies this station as roadside. Station Leipzig Mitte (Leipzig Center, LC) provides $PM_{10}$ and eBC measurements. It is located southwest of the Leipzig main railway station at a junction of the inner-city ring, which is a multi-lane road (ADTC of 47600 vehicles). Therefore, also this

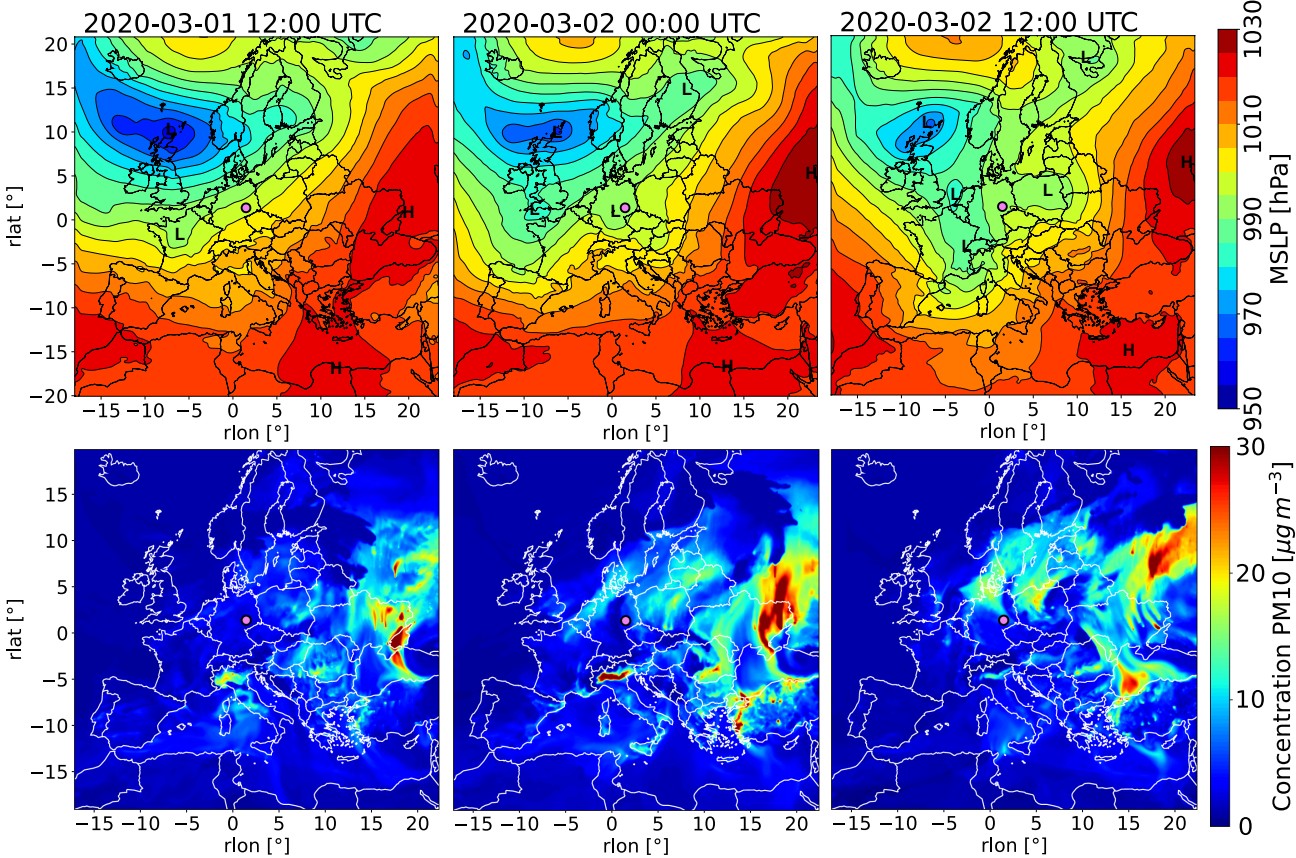

**Figure 1.** Overview of the synoptic-scale transport and air quality patterns over Europe during the simulation period in terms of maps of mean-sea level pressure (a-c), and modeled near-surface $PM_{10}$ concentrations (d-f). The magenta dot marks the location of the city of Leipzig in eastern Germany. The maps are in rotated geographic coordinates, where the coordinates of the rotated pole are 40°N, -170°E (see Sect. 2.3 for a detailed description of the model setups).

station is classified as roadside. Compared to the site LL, there is more open area around the station LC ($f_{sky} = 0.79$), with the closest significant building (height × width of $27\,\text{m} \times 50\,\text{m}$) being to the south of the station at a distance of approximately $45\,\text{m}$. Furthermore, due to a local park adjoining to the east, the influence of traffic emissions at the measurement site can be expected to vary according to the prevailing wind direction. Station Leipzig West (LW), located within the western outskirts of Leipzig inside a park, is a background station for $PM_{10}$, as it is also secluded by lines of trees from a nearby road (ADTC of 8600 vehicles). Station Leipzig Eisenbahnstr. (LE) has a long history as a scientific measurement site and is thus well documented from previous air-quality studies (see e.g. Klose et al. 2009, Wiesner et al. 2021). The measurement equipment is located next to a window on the third floor of an apartment house (inlet height is approx. $6\,\text{m}$ above the road) flanking

a frequently traffic-congested street canyon (ADTC of 11500 vehicles). The cross section of the street canyon is symmetric ($20\,\mathrm{m} \times 20\,\mathrm{m}$). Regularly occurring crossroads divide the street canyon into segments of $70\,\mathrm{m} - 110\,\mathrm{m}$ length, with the closest crossroad (ADTC of 11800 vehicles) being to the west at a horizontal distance of about $35\,\mathrm{m}$ from the measurement site. While
this side is also classified as roadside, its inlet position high above the road makes it more representative to the average concentrations within the street canyon. Depending on the development of the street-canyon vortex, however, it may be also more directly exposed to high pollution concentrations. Finally, the side Leipzig TROPOS (LT) is a background station for eBC, as it is located on the roof top of the TROPOS institute's building at a height of $16\,\mathrm{m}$ and at a distance of at least $100\,\mathrm{m}$ from any busy roads.

PM$_{10}$ concentrations are directly and near-continuously measured using the tapered element oscillating microbalance (TEOM) system (scientific ambient particulate monitor TEOM 1405, Thermo Fisher Scientific Inc.). TEOM derives PM mass concentrations from the frequency-change of an oscillating hollow tube caused by deposited material at one end of the tube (Page et al., 2007). Real-time measurements are averaged to hourly-mean values with a stated precision of $\pm 2.0\,\mathrm{\mu g}$ and an accuracy of $0.75\,\%$ (TFS, 2019). eBC is indirectly retrieved from optical principles with multi-angle absorption photometers (MAAP
5012, Thermo Fisher Scientific Inc.). MAAP estimated the absorption coefficient of an aerosol probe from the transmission and back-scattering of light at a wavelength of $637\,\mathrm{nm}$, where eBC is the main absorber (Petzold and Schönlinner, 2004). The eBC mass concentrations calculated with a mass absorption cross section of $6.6\,\mathrm{m^2\,g^{-1}}$ are assumed to be directly comparable with modeled BC mass concentrations, and have an uncertainty between $5\,\%$ and $12\,\%$ according to different sources (Wiesner et al., 2021).

**Table 1.** Overview of the air-monitoring sites used for model evaluation. For the pollutants, only the relevant species to our model evaluation are denoted. Note also, that the station LC is both a public and scientific site.

| Location | Lützner Str. | Richard Wagner Str. | Schönauer Str. | Eisenbahnstr. | TROPOS |
|---|---|---|---|---|---|
| **Label** | LL | LC | LW | LE | LT |
| **Characteristics** | street canyon W-E | multi-lane road | park | street canyon W-E | roof top |
| **Classification** | roadside | roadside | urban background | roadside | urban background |
| **Coordinates °E, °N** | 12.335, 51.336 | 12.377, 51.344 | 12.298, 51.318 | 12.401, 51.346 | 12.434, 51.353 |
| **Inlet height** | $1.7\,\mathrm{m}$ | $4\,\mathrm{m}$ | $4\,\mathrm{m}$ | $6\,\mathrm{m}$ | $16\,\mathrm{m}$ |
| **Pollutants** | PM$_{10}$ | PM$_{10}$, BC | PM$_{10}$ | BC | BC |

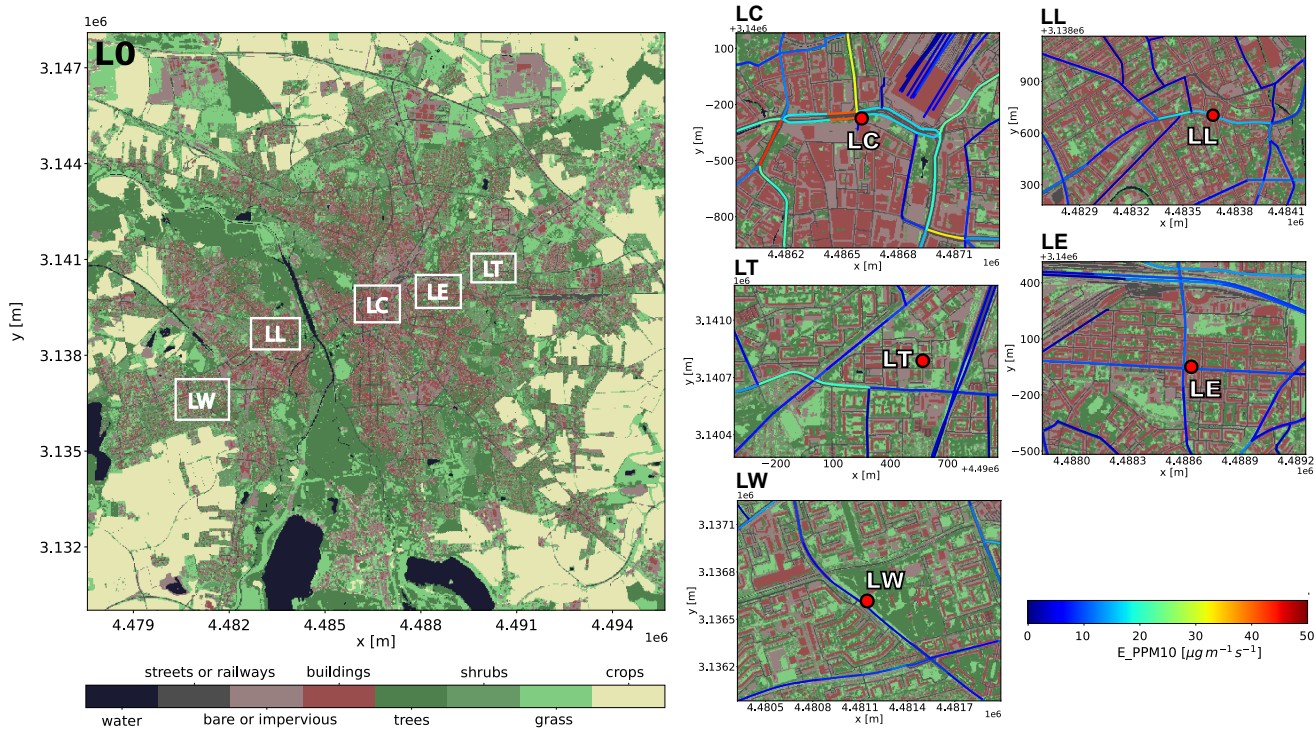

**Figure 2.** Map of the city area of Leipzig, which is also selected as the coarse-grid CAIRDIO simulation domain L0 introduced in Section 2.4.1. Each of the white boxes contains an operational air monitoring site used for model evaluation. In addition, a magnified view of the area within each box shows the local environment around the corresponding air-monitoring site, which is highlighted by a red circle. These areas also correspond to the CAIRDIO subdomains introduced in Section 2.5. Traffic-PPM$_{10}$ emissions of major roads are represented by line sources.

### 2.2.2   In-situ and remote sensing meteorological observations

The evolution of the PBL has an important influence on the distribution and levels of urban air pollutants. To evaluate the properties and the vertical structure of the simulated PBL, a comprehensive set of in-situ and remote sensing measurements has been used.

At the TROPOS institutes site, lidar-based remote sensing instruments are routinely deployed to monitor aerosol composition and dynamics within the PBL and also above. With the portable Raman lidar Polly-XT (Engelmann et al., 2016) as part of the ACTRIS subnetwork PollyNET (Baars et al., 2016), vertical profiles of aerosol optical properties are measured continuously. From the vertical gradient in the profiles of the attenuated backscatter coefficient, the MLH can be estimated (Baars et al., 2008). While this method is reliable for the daytime with an aerosol-loaded PBL and a clean free troposphere above, with a well-differentiated aerosol layer, the vertical contrast in the backscatter profiles during night time is often much lower. The

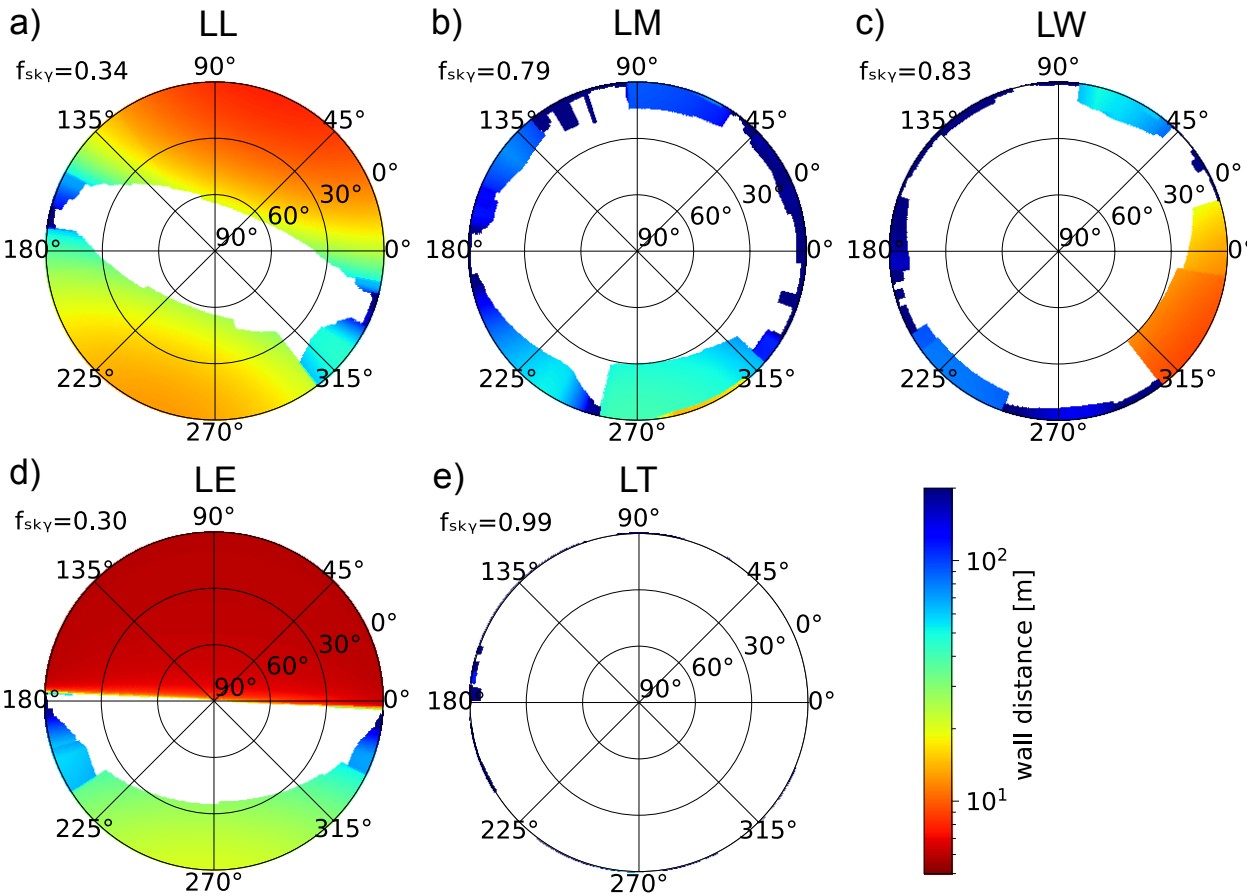

**Figure 3.** Simulated view on surrounding buildings in spherical coordinates at the exact 3-D locations of the inlets for the stations (a) LL, (b) LC, (c) LW, (d) LE, and (e) LT. The colors indicate the distance of instrument inlets to building walls, while visible sky is shaded in white. Additionally, the sky-view factors $f_{sky}$ are computed as the fraction of the solid angle of the hemisphere not blocked by buildings.

backscatter gradient from the nocturnal boundary layer to the residual layer (the remaining aerosol layer from daytime) is often week and the lower-height detection threshold of the lidar system (overlap issue, e.g. Wandinger and Ansmann 2002) decreases the confidence of the method then and often prohibits the determination of the nocturnal MLH. Besides the Polly XT lidar, a HALO photonics streamline XR Doppler lidar was also operated at the TROPOS site during the study period. n vertical staring mode (for dertmination of vertical air motions) but also in scanning mode (PPI) for the determination of the horizontal wind velocity. In vertical staring mode, this lidar can be used to accurately observe vertical motions (uncertainty of less than $0.1\,\mathrm{m\,s^{-1}}$) in atmospheric regions with significant aerosol load (PBL, lifted aerosol layers, clouds) (Bühl et al., 2015). In scanning mode (PPI), it also allows to retrieve vertical profiles of the horizontal wind by the same Doppler principle.

Respective profiles were obtained with a frequency of $10\,\text{min}$ throughout the simulation period. Data points with a relative uncertainty in one of the two wind components of larger than $20\,\%$ are discarded. The estimation of the night-time MLH is based on the standard deviation of the observed vertical motions as indicator for turbulence: To make a single estimate, starting from the surface, the control volume of a box containing measurements ($\Delta t \times \Delta z$) is vertically increased until the computed standard deviation of contained measurements falls below a given threshold. The MLH is then determined from the height of the control volume. Results are sensitive to the selected time increment, the vertical resolution and the standard deviation threshold. Schween et al. (2014) give a relative change of the estimated MLH by $15\,\%$ for a variation of the threshold by $25\,\%$. In addition to the MLH estimation, the thermal stratification of the boundary layer can be mapped using in-situ measurements from atmospheric soundings performed at the DWD meteorological observatories Lindenberg ($150\,\text{km}$ northeast of Leipzig, every 6 hours) and Meiningen ($160\,\text{km}$ southwest of Leipzig, every 12 hours). While these data cannot be used to evaluate the local city simulations, they can nevertheless be used to evaluate the coarser-scale meteorological simulations for central Germany.

In addition to the air-quality measurements, hourly-averaged observations of wind speed, wind direction and air temperature from the sites LL and LC, respectively, are used to evaluate the urban wind field and urban heat island effect. This comparison can contribute to the confidence and discussion of the conducted dispersion simulations. To complete the set of surface observations, hourly precipitation totals from two meteorological sites operated by the German weather service (Deutscher Wetterdienst, DWD) are used to evaluate the model representation of precipitation and its impact on air quality. The first site Leipzig/Halle is located $14\,\text{km}$ northwest from the Leipzig city centre, and the second site Leipzig/Holzhausen is located $5.6\,\text{km}$ southeast from the Leipzig city centre.

## 2.3 Mesoscale air-quality modeling

As air pollution is not only influenced by local processes, all relevant larger scale sources and transport have to be considered in the city-focused, urban microscale simulations in terms of boundary conditions. Such a multiscale approach requires tailored model setups with a scale-appropriate prioritization of the dominating processes. Besides the long-range transport, physico-chemical reactions contributing to significant secondary particulate matter (SPM) formation have to be considered on the continental and regional scales, for which in this study the online-coupled mesoscale CTM COSMO-MUSCAT (Wolke et al., 2012) is employed. COSMO-MUSCAT uses the regional model COSMO (Doms and Baldauf, 2018) as the meteorological driver, which was maintained and operationally used by the DWD until recently. Important multi-phase reactions in MUSCAT leading to SPM involve the gaseous compounds ammonia, nitric acid, and sulfuric acid, which themselves are important air pollutants. Additionally, seasonally dependent secondary organic aerosol (SOA) formation is included in the set of chemical reactions composed by the mechanism RACM-MIM2 (Stockwell et al., 1997; Taraborrelli et al., 2009). The remaining fraction of $PM_{10}$ is primarily emitted (PPM), and approximated by chemically inert tracers that are only subjected to physical atmospheric removal processes. Figure 4 gives an overview of the bulk $PM_{10}$ decomposition in MUSCAT as it is available from the model output. COSMO-MUSCAT is applied on a hierarchy of refined domains, with a one-way nesting technique providing the boundary condition for each consecutive simulation (see Figure 5 for an overview of the domains,

which are referred to hereafter with "M<number>"). This model setup has already been used to provide quasi-operational air-quality forecasts for the citizen-science campaign WTImpact (Heinold et al., 2019; Tõnisson et al., 2021). The outermost domain M0 has a horizontal resolution of $14\,\mathrm{km}$ and covers entire Europe. This domain is initialized and driven by re-analysis data from the global meteorological model ICON (Zängl et al., 2015), which is operationally run by DWD. Initialization and boundary conditions for air chemistry are interpolated from operational air-quality forecasts with the model system ECMWF IFS (Copernicus Atmosphere Monitoring Service) (Flemming et al., 2015). The domain M0 is simulated for an extended period in time (at least two weeks) ahead of the actual simulation period of two days. This allows for a proper relaxation of the initial distribution of air-chemistry constituents to the new model setup, as a different meteorological model, air-chemistry mechanism, and emission dataset are used. Simulation results for air chemistry are interpolated on domain M1 with $2.2\,\mathrm{km}$ resolution covering middle/eastern Germany, part of Czech Republic and Poland. For more accurate meteorological boundary conditions, re-analysis data from the operationally run COSMO-D2 model is used instead of the meterological output from domain M0. Output from domain M1 is interpolated on domain M2 with $1.1\,\mathrm{km}$ resolution. Simulation results from domain M2 are finally interpolated on the innermost domain M3 near the lower end of the mesoscale with $550\,\mathrm{m}$ horizontal resolution containing the city of Leipzig in its center. At this scale, the influence of the urban canopy and building environment is already considered based on the double-canyon effect parameterization (DCEP) by (Schubert et al., 2012). DCEP includes 3 different types of urban canopy elements (ground, wall and roof elements), which are configured in idealized double-canyon segments. A preprocessor is used to derive horizontal coverage of these segments in each grid cell, as well as probabilistic and geometric parameters of the canopy elements using a detailed building geometry dataset available for entire Saxony. DCEP computes surface fluxes for momentum, heat and turbulent kinetic energy (TKE), as well as solves the equations for radiative transfer and heat balance of the canopy elements. From the latter mean temperatures are available too.

The emissions used in our modeling study incorporate several source categories, which are based on the Selected Nomenclature for Air Pollution (SNAP). In detail, these include emissions from combustion in the production and transformation of energy (SNAP1), non-industrial combustion plants (SNAP2), industrial combustion plants (SNAP3), industrial processes without combustion (SNAP4), extraction and distribution of fossil fuels and geothermal energy (SNAP5), use of solvents and other products (SNAP6), road transport (SNAP7), other mobile sources (SNAP8), waste treatment and disposal (SNAP9), and agriculture (SNAP10). The large range of spatial coverage and horizontal grid spacing addressed with the nested mesoscale model domains requires the use of different appropriate emission datasets. For domain M0, emissions are interepolated from the TNO-MACC2 dataset of the year 2009 (Kuenen et al., 2014). For the domains M1 and M2, German-wide area emissions are provided by the German Environment Agency (Umweltbundesamt, UBA) for the year 2015 with $1\,\mathrm{km}$ spatial resolution, while emissions for the Czech Republic and Poland are again based on the aforementioned TNO-MACC2 emissions. The emission datasets for domain M3 and the consecutive CAIRDIO domains are primarily based on better resolved raster emissions (uniform $500\,\mathrm{m}$ grid spacing) provided by UBA for parts of Eastern Germany. Nevertheless, traffic emissions within the city margins of Leipzig were carefully substituted with a gridded line-source database valid for the year 2016 and provided by the LfULG. All of the aforementioned emission datasets are static in time, i.e. they represent annual mean emission rates per unit area, respective unit length. For time-depended emissions, the aforementioned static datasets are modulated with time profiles

specific to each SNAP category (available from the TNO-MACC2 database). The time profiles incorporates different temporal
scales, including monthly, weekly and diurnal changes. The temporal resolution is 1 h, with linear interpolation applied be-

tween clock hours. The spatially integrated and temporally modulated emissions are shown in Fig. 6 for the simulation period.
Accordingly, the road transport category is by far the most important contributor to BC emissions, followed by other mobile
sources and non-industrial combustion.

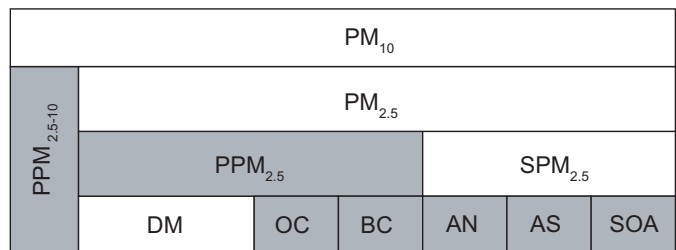

**Figure 4.** PM decomposition in the mesoscale CTM COSMO-MUSCAT: $PM_{10}$ is the bulk mass concentration of all particles with mean
diameter $d < 10\,\mu m$. $PM_{10}$ is further composed into the two size fractions $PPM_{2.5-10}$ (primary matter) and $PM_{2.5}$. $PM_{2.5}$ again contains
primary contributions from BC, organic carbon (OC), and dust and metallic particles (DM). The secondary fraction is mainly composed
of ammonium nitrate (AN), ammonium sulfate (AS) and secondary-aerosol (SOA) particles. Boxes colored in gray indicate direct model
outputs.

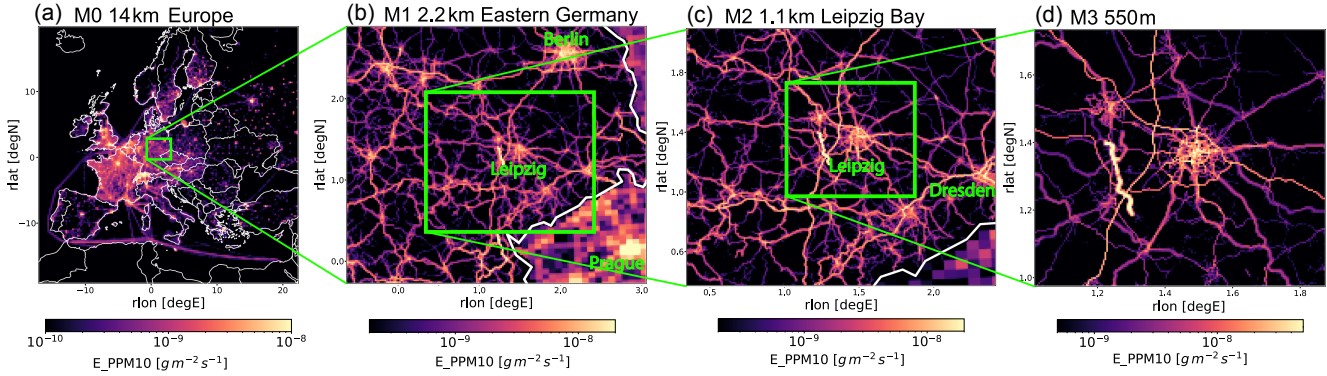

**Figure 5.** Nested domains of the precursor simulations using COSMO-MUSCAT: (a) M0, 14 km, (b) M1, 2.2 km, (c) M0, 1.1 km, and (d)
M3, 550 m. Displayed are the primary $PM_{10}$ emissions of the transport sector (SNAP categories 7 and 8). Note that the displayed range
differs between the plots to account for the increased resolution of road emissions from (a) to (d). The maps are in rotated geographic
coordinates, where the coordinates of the rotated pole are 40°N, -170°E.

## 2.4 Intra-urban scale dispersion modeling

### 2.4.1 City-wide CAIRDIO model setup

The intra-urban scale is addressed with the large-eddy simulation-based dispersion model CAIRDIO. This model solves the prognostic fluid dynamic equations either in Boussinesq or anelastic approximation. Obstacles on the surface are implemented by diffuse obstacle boundaries (DOB), while the geographic topography itself is represented by terrain-following coordinates. DOB modify the effective geometric properties of grid cells, like their volumes and face areas, so that they directly impact the conservation of momentum in the semi-discrete formulation of the governing equations by a finite volume method. The

advantage of DOB is that buildings can be represented across a wide range of spatial resolutions, including non-eddy-resolving resolutions where the grid spacing is in the range of building size (a few tens of meters) or even larger, and a major part of turbulent mixing is parameterized. Note that the current model version v2.0 does not yet include the sophisticated multi-phase air chemistry employed in the host simulations with MUSCAT. Therefore, pollutant concentrations are simply governed by advective, diffusive and deposition tendencies. For slowly reacting or inert pollutants, like PM, the non-reactive treatment is

a valid approximation when considering the limited domain extend and consequential short retention time of pollutants. For a more detailed description of CAIRDIO v1.0, the reader is referred to Weger et al. (2021). In appendix A improvements of the actually used model version v2.0 are listed.

The local urban domains of the CAIRDIO model are hereafter denoted by "L<number>". For the city-wide simulation, domain L0 is horizontally resolved with uniform $40\,\mathrm{m}$ grid spacing. This resolution showed to be satisfactorily accurate in a

first model validation study based on wind-tunnel data also presented in Weger et al. (2021). For DOB, the same geometric building dataset as already used for deriving the parameters in DCEP is used. At the horizontal scale of $40\,\mathrm{m}$, buildings are effectively represented as diffuse obstacles. As a consequence, the subgrid-scale mixing length is of similar magnitude or even larger than the average space between buildings. This implies that the simulation is mostly non-eddy resolving (similar to a RANS approach) within the urban canopy. For a more accurate representation of vertical gradients and mixing processes near

the surface, the vertical dimension is kept much finer resolved, with $5\,\mathrm{m}$ grid spacing within the urban canopy. Increased grid stretching is applied above, with a maximum stretching factor of 4 near the domain top. Note that vertical resolution is not as computationally expensive as horizontal resolution, because the Courant-Friedrichs-Lewy (CFL) criterion, which limits the explicit integration time step, is mostly determined by the horizontal wind speed. The horizontal coverage of domain L0 is roughly $18\,\mathrm{km} \times 18\,\mathrm{km}$, which is extensive enough to not only accommodate the complete city of Leipzig but also a sufficient

fetch to allow for a relaxation of the lateral boundary conditions. The domain top is located at about $350\,\mathrm{m}$ height, which is generally below the vertical extend of a typical convective PBL. However, a suitable boundary condition allows for vertical motions exiting/entering at the domain top, which is further explained in Appendix A0.3. While the surface orography within domain L0 is not mountainous, subtle effects from it can still influence meteorology. CAIRDIO can be used with terrain-following coordinates, which in this simulation are inferred from surface elevation data (DGM1) provided by the Staatsbetrieb

für Geobasisinformation und Vermessung Sachsen (GeoSN). This dataset with a spatial resolution of $1\,\mathrm{m}$ has also been corrected for vegetation and buildings and is thus compatible with the explicit representation of buildings by DOB. Surfaces fluxes

of momentum, heat and moisture from vegetation, as well as other types of land cover (lakes, bare soil, subgrid-scale structure of buildings) are parameterized using Monin–Obukhov similarity theory (MOST). Note that this includes also urban trees, which are therefore most simply represented by the surface-roughness approach (see e.g. Salim et al. 2015 for a discussion of the limitations). In MOST, each surface type is characterized by a parametric roughness length $z_0$. Table 2 lists the $z_0$ values related to each land-cover class used in the model. Land-cover is based on a combination of the Pan-European land cover map for 2015 with $30\,\mathrm{m}$ spatial resolution (Pflugmacher et al., 2018), and the more detailed land-cover map by Banzhaf and Kollai (2018) (better than $5\,\mathrm{m}$ resolution) for most of the urban area. The combined dataset is depicted in Fig. 2 for domain L0, as well as for the finer resolved nested subdomains introduced in Section 2.5 addressing the LES-to-LES nesting.

The emissions used for domain L0 are on the same basis as the ones already used for mesoscale domain M3, which are, the UBA area emissions for industry (SNAP 3, SNAP 4, SNAP 6, SNAP 9) and residential combustion (SNAP 2) with $500\,\mathrm{m}$ resolution, and Leipzig traffic emissions represented by line strings (SNAP 7 and partly SNAP 8). The latter emissions (also including the railway network) are simply gridded on domain L0 without loss of spatial accuracy. In this regard, it can be noted that the spatial accuracy of both the line-string representation of emissions and polygon representations of buildings are sufficient such that intersections of different geometries are avoided. The area emissions for industry and residential combustion are refined by the following method: Based on the classification of buildings as commercial or residential, the area emissions are firstly allocated to the respective building sites included in a $500\,\mathrm{m} \times 500\,\mathrm{m}$ coarse raster cell, whereas the fractional building volumes are taken as weighting factors. In a second step, the building-accumulated emissions are gridded on the finer domain L0. The effective emission height is computed from the gridded average building height. The advantage of this emission downscaling is currently investigated in a separate sensitivity study. The time modulation of the static emissions uses the same temporal profile as already used for the mesoscale simulation setups.

**Table 2.** Roughness length values $z_0$ of the different land-cover classes used in all CAIRDIO simulations.

| Land-use class | $z_0$ |
| --- | --- |
| Water surface | $0.001\,\mathrm{m}$ |
| Bare soil | $0.01\,\mathrm{m}$ |
| Grass land | $0.03\,\mathrm{m}$ |
| Crop land | $0.07\,\mathrm{m}$ |
| Shrubs | $0.3\,\mathrm{m}$ |
| Trees | $1.0\,\mathrm{m}$ |
| Streets and railways | $0.1\,\mathrm{m}$ |
| Buildings | $0.2\,\mathrm{m}$ |
| Other impervious surface | $0.05\,\mathrm{m}$ |

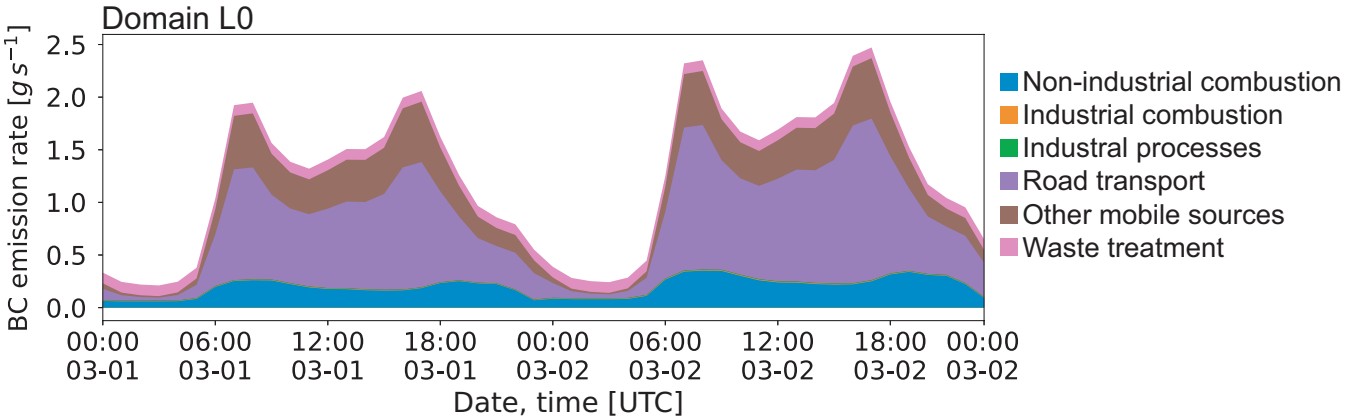

**Figure 6.** BC emission rate (TNO-MACC2 time profile) over simulation time accumulated over the domain L0 subdivided into the relevant SNAP categories.

### 2.4.2 Mesoscale forcings and boundary conditions

Simulation results from COSMO-MUSCAT mesoscale domain M3 are used to drive the meteorological and air-pollution fields of the city-scale domain L0. Initial and boundary conditions for the meteorological prognostic fields, which include the 3-D
wind components, potential temperature $\Theta$, specific humidity $Q_V$, and subgrid-scale TKE $E_{sgs}$, are spatially interpolated using tricubic interpolation. Note that tricubic interpolation preserves spatial details better than trilinear interpolation, but is not well-suited for positive scalar fields featuring large gradients. Therefore, trilinear interpolation is used for the air-pollution fields. The 3-D interpolation procedure is carried out as a sequence of 2-D horizontal interpolation followed by vertical interpolation. For the horizontal interpolation, the Climate Data Operators (CDO) software (Schulzweida, 2019) is used, which is convenient
to remap data from rotated lat/lon coordinates of the COSMO-MUSCAT model directly to Lambertian azimuthal equal-area coordinates (epsg:3035) of the L0 grid. Vertical interpolation is based on the 3-D height of half levels (HHL), which coincide with the locations of vertical velocity of the staggered grid. After horizontal remapping of the HHL field of the M3 grid, the vertical interpolation weights are generated by computing Lagrange polynomials of the desired accuracy from the HHL data. As the CAIRDIO simulation employs a finer grid spacing near the ground than COSMO-MUSCAT, the first vertical levels
need to be extrapolated, for which a level with zero-height is introduced. At this zero-height level, all wind components are set to zero, and the potential temperature as well as specific humidity fields assume the respective surface values $\Theta^S$ and $Q_V^S$. For the air-pollution fields, constant values from the first MUSCAT layer are extrapolated.

Lateral boundary conditions for the prognostic subgrid-scale TKE equation in the microscale model CAIRDIO are derived by applying a scale separation on the spatially interpolated subgrid-scale TKE of the COSMO-MUSCAT simulation (denoted
by coarse) $E_{sgs}^c$. $E_{sgs}^c$ is split into a part $E_{res}^f$ that can be resolved on the CAIRDIO L0 grid (denoted by fine) and a still unresolvable component $E_{sgs}^f$. The energy splitting can be approximated by integrating the well-known Kolmogorov spectrum

for the inertial subrange $E(k) \propto k^{-5/3}$ up to the different cut-off wave-numbers $k_{min}$ of the fine and coarse grids. $k_{min}$ can be directly related to the subgrid-mixing scale $\Delta_{sgs}$, then the following expression follows:

$$E_{sgs}^f = E_{sgs}^c \left( \frac{\Delta_{sgs}^f}{\Delta_{sgs}^c} \right)^{2/3}. \tag{1}$$

$\Delta_{sgs}^f$ can be crudely approximated by twice the horizontal grid spacing (corresponding to the Nyquist wavenumber). Note that the horizontal grid spacing in typical PBL simulations is equal or larger than the vertical grid spacing and is thus the dominant cut-off scale. $\Delta_{sgs}^c$, on the other hand, can be related to the master-mixing length of the mesoscale simulation. $E_{sgs}^f$ is finally the lateral boundary condition for the prognostic subgrid-scale TKE equation, which determines the eddy diffusivities. The lateral boundary condition for the 3-D wind vector is a Dirichlet/radiation condition that can flexibly distinguish inflow

from outflow regions. For inflow regions, a superposition of the interpolated mesoscale wind field and recycled turbulence is prescribed. The scale-separation applies in a similar way for the turbulence recycling scheme, as the cut-off wavelength of the extraction filter is chosen similar to $\Delta_{sgs}^c$. Consequently, the recycled turbulent fluctuations are scaled with the resolvable energy part $E_{res}^f$. At the domain top, a special boundary condition for velocity, which is quite similar to the turbulence recycling scheme for the lateral boundaries, is used. This boundary condition allows for a simultaneous prescription of the external

mesoscale fields and small-scale turbulent motions reaching/extending beyond the domain top. Further details are addressed in Appendix A0.3. As the turbulence recycling scheme can only extract and amplify existing turbulence, the potential temperature is disturbed by the cell-perturbation method of Muñoz-Esparza et al. (2015) across the full domain at model initialization.

Besides the initial and boundary conditions for the prognostic fields, another important forcing includes the heat and moisture fluxes from the land surface. For their parameterization in CAIRDIO, surface potential temperature $\Theta^S$ and surface specific

humidity $Q_V^S$ are needed. Instead of employing an own land-surface model in CAIRDIO, a simpler approach is used in this study, which consists of a downscaling of the respective prognostic surface fields from the mesoscale simulation M3. This can be referred to as a one-way coupling with the land surface, in contrast to a fully two-way coupling, which also considers the online feedback of the atmosphere on the land surface. A drawback of this neglected feedback is that the land-surface variables inherently lack the dynamic spatial variability at the finer scales that are only represented in the LES model. This

may adversely impact the computed surface fluxes and as a final consequence the thermal stratification of the boundary layer. Also it is to point out that small-scale radiative effects, like partial shading inside a street canyon cannot be represented by this simple approach. Nevertheless, these potential sources of modeling error are accepted in favor of a simple solution, until an own land-surface for CAIRDIO is developed. The used down-scaling method essentially is a linear regression model that is based on the assumption of land cover being the most significant co-determinant of the small scale variability. This assumption

applies only for limited horizontal domain extents with non-mountainous terrain, which, however, is quite well satisfied with domain L0. For application outside these limits, the approach can be easily extended to a multiple linear regression model in order to consider other important explanatory variables, like, e.g., surface height, or the influence on horizontal position, which can be approximated by a bilinear function.

Explaining the method on the basis of surface potential temperature, in a first step, the mesoscale field is decomposed into

a filtered or mean state $\overline{\Theta^S}$ and a fluctuating part $\Theta^{S\prime}$. Additionally given are the different land-cover fractions on the same mesoscale grid. These fractions are put in a $m \times n$ matrix $\mathbf{L}$, whereas the dimension $m$ states for the number of independent land-cover classes considered, and dimension $n$ for the number of horizontal grid cells. Then, it is possible to solve for the unknown land-cover related potential temperature fluctuations $\Theta^{S\prime}_{Lc}$ by minimization of the following least-square problem:

$$||\mathbf{L}\Theta^{S\prime}_{Lc} + b - \Theta^{S\prime}|| \; =! min. \tag{2}$$

The vector $b$ contains the potential temperature contribution from a-priori determined land-cover classes, which are already known for lakes and urban surfaces, the latter being direct outputs of the urban parameterization DCEP in COSMO. For more robust fitting results, we consider only forests, open vegetation (includes grass land, shrubs, and crops) and bare soil (minus the fraction of urban surfaces) as independent classes. After solving Eq. 2, the high-resolution field of domain L0 is composed by multiplying the obtained land-cover dependent fluctuations $\Theta^{S\prime}_{Lc}$ (including $b$) with respective land-cover

fractions given for domain L0 and addition to the horizontal mean state, respective horizontally interpolated filtered state $\overline{\Theta^S}$. Because CAIRDIO uses a 3-D building structure, potential temperature values of these elevated horizontal and vertical surfaces are still missing. These values can, however, be directly inferred from the additional output fields of the mean roof and wall temperature computed with the DCEP parameterization. $Q^S_V$ is fitted in a similar fashion like $\Theta^S$, with a further simplification that $Q^S_V$ is set to zero for the impervious surface fraction, thus rain evaporation is neglected. Fig. 7 qualitatively shows the

described downscaling approach based on an exemplary $\Theta^S$ field for 2 March 2020, 12:00 UTC, when sunshine prevailed. For the resulting reconstructed field of domain L0, a top-down projection is shown. The sun-lit roofs are the warmest surfaces with a quite even distribution due to a prescribed constant roof albedo of 0.16 in DCEP. Considerably cooler are the ground surfaces inside the partly shaded street canyons, followed by the forest areas, and finally by the seasonally cold lakes with the lowest surface potential temperature. While in the given example, surface orography is mostly flat and the surface elevation could

be neglected as an additional disturbing factor, this approach may be an oversimplification for areas with more mountainous terrain. For a limited quantitative evaluation of the downscaling method in the framework of this case study, we refer to the comparison of modeled near-surface air temperature with observations in Appendix B (Fig. B1e-f).

## 2.5    One-way LES to LES nesting

     In order to quantify the influence of spatial resolution on results of the city-scale CAIRDIO simulations, spatially limited sub-

domains LW, LL, LC, LE, and LT, each of them centered around an air monitoring site, are offline-nested into the parent domain L0. For these local domains, the horizontal grid spacing is gradually decreased from $40\,\mathrm{m}$ to $5\,\mathrm{m}$. The finest grid spacing permits conventional building-resolved simulations. To drive such nested simulations, prognostic fields from the CAIRDIO domain L0 are available in $30\,\mathrm{s}$ intervals. The horizontal interpolation is carried out in the same way as already explained for the mesoscale forcing fields in Section 2.4.2. Since both the parent and nested domains use the same vertical grid, vertical interpolation is not

needed in this case. The lateral boundary conditions for the subgrid-scale TKE equation are again based on a scale splitting according to Eq. 1. Therefore, the coarse-grid subgrid-scale energy from the CAIRDIO L0 grid is used now. For the resolved

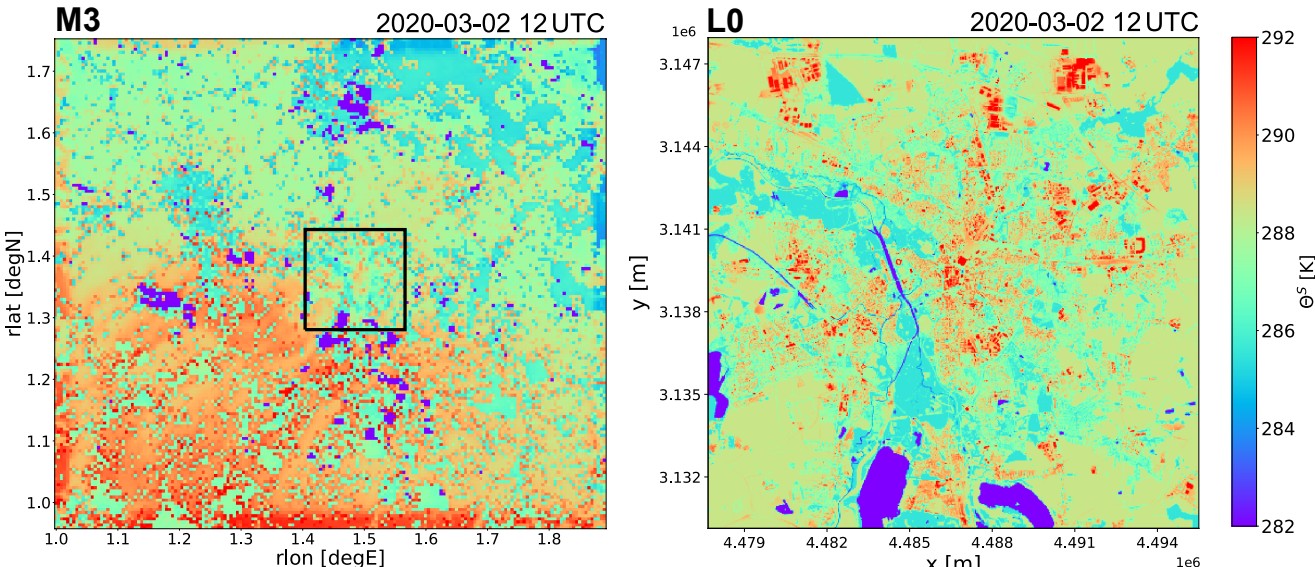

**Figure 7.** Diagnostic $\Theta^S$ downscaling from the mesoscale COSMO-MUSCAT domain M3 to the urban-microscale CAIRDIO domain L0 as by the explained land-cover method in Sect. 2.4.2.

part $E_{res}^f$, which scales the inserted turbulent fluctuations at the inflow boundaries, a slightly modified formula is used in order to consider the missing contribution of numerical diffusion in $E_{sgs}^c$:

$$E_{res}^f = E_{sgs}^c \left[ \left( \frac{\lambda_{cut}}{\Delta_{sgs}^c} \right)^{2/3} - \left( \frac{\Delta_{sgs}^f}{\Delta_{sgs}^c} \right)^{2/3} \right], \quad (3)$$

where the extraction-filter width is set to $\lambda_{cut} = 140\mathrm{m}$, which is considerably larger than $\Delta_{sgs}^c = 80\,\mathrm{m}$. Note that the use of an exponential filter function results in a smooth cut-off range, with $\lambda_{cut}$ defined as the wavelength that is scaled $e^{-1}$-fold.

In Figure 8 the described nesting method with the energy-scale separation is demonstrated by an example with domain LE and $5\,\mathrm{m}$ horizontal grid spacing, which features a stably-stratified, shear-driven PBL. The plots of the dominant velocity component $v$ (Fig. 8a and b) show that well-developed turbulence already exists near the southern inflow boundary, which qualitatively

matches the turbulence more distant from the boundary well. This is also quantitatively shown by the derived energy spectra shown for the x-dimension (Fig. 8c), which do not evolve much when moving further away from the inflow boundary. Note that the inertial subrange is followed by the dissipation range, which can be attributed to the combined (dissipative and dispersive) numerical error of the advection scheme at significant convective speeds (Yalla et al., 2021).

The surface fields $\Theta^S$ and $Q_V^S$ are again obtained from the corresponding mesoscale fields using the land-cover based method

described in Sect. 2.4.2. An additional scaling is applied, such that the computed horizontally averaged values are independent of the spatial resolution and also correspond to the average values of the congruent part of the parent domain.

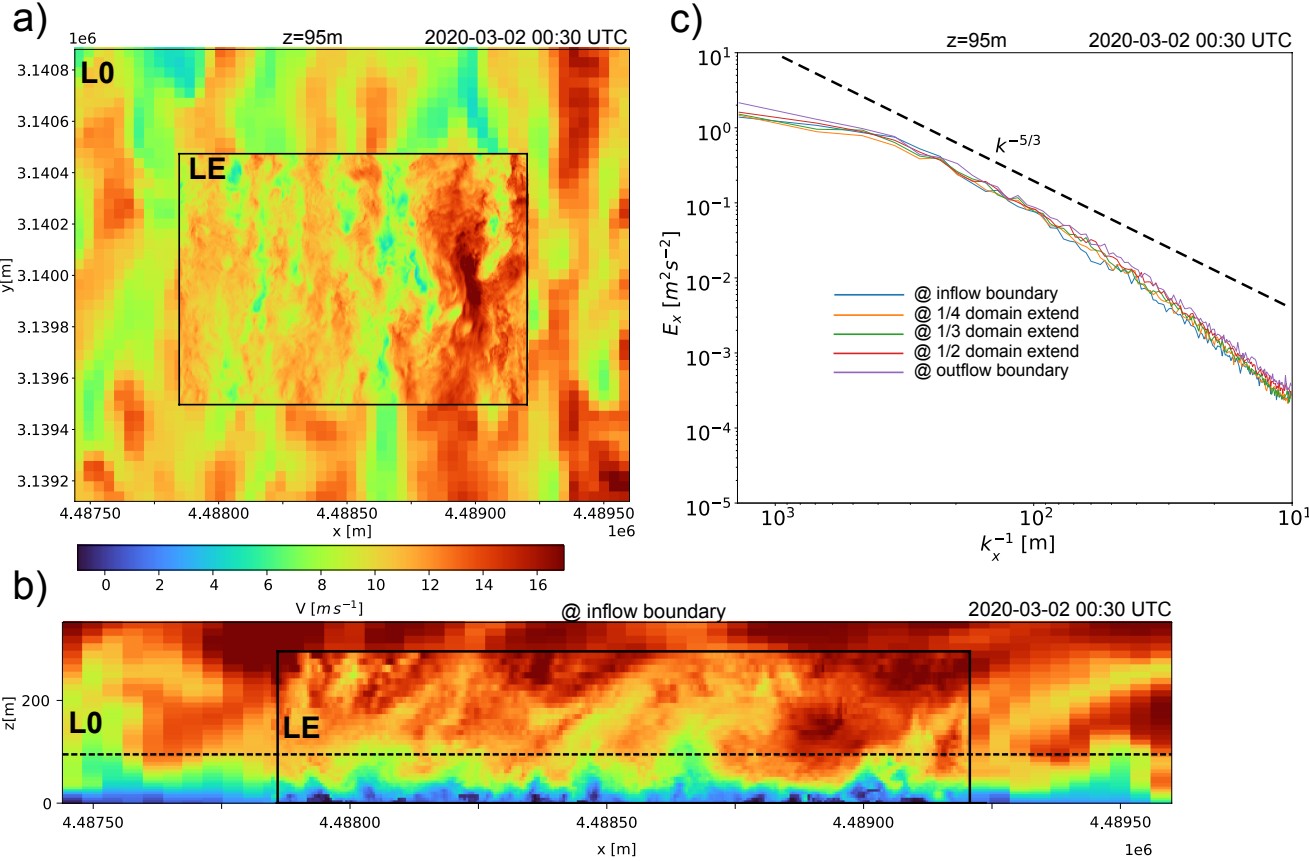

**Figure 8.** Depiction of a turbulent PBL flow simulated with the parent CAIRDIO domain L0 (40 m) and the offline-nested sub-domain LE (5 m) with a southern inflow boundary. In panel (a) the velocity component $v$ is shown at 95 m height. Small-scale turbulence extracted from domain LE is superimposed on the interpolated coarse-grid wind field at the boundary-ghost cells, resulting in well-resolved turbulent structures at the inflow boundary as shown in (b) for the vertical plane. The horizontal dashed line in (b) marks the position of the horizontal plane shown in (a). In (c), spectra of resolved turbulent kinetic energy are shown for the x-axis at 95 m height and 5 m horizontal grid spacing. Plotted are, the energy spectra at various positions along the y-axis.

## 3   Results

### 3.1   Synoptic and urban planetary-boundary layer meteorology

In the following, an overview of the meteorological conditions and resulting PBL characteristics during the simulation period from 1 March 2020, 00:00 UTC to 3 March 2020, 00:00 UTC is given. The implications of the variable PBL structure on the modeled concentrations and transport of local air pollution are then qualitatively discussed based on model outputs from the CAIRDIO domain L0. As already briefly mentioned in Section 2, the weather in Leipzig during the simulation period was

influenced by a low pressure system. The large pressure gradients ahead of the troughs (see again Fig. 1a-c) imply windy conditions over Leipzig for most of the time as it is shown by both model and lidar-based observational data in Fig. 11a.

Indeed, the plotted wind barbs indicate the presence of a pronounced shear layer during the first $12\,h$ of the simulation and just before the passage of the first trough on 2 March around 00:00 UTC, when southwesterly winds reached up to $75\,km\,h^{-1}$ above $700\,m$ height. Note that while both model and measurements agree well in that maximum value, the decrease in wind speed towards the surface is more gradual in the model. This leads to an overestimation of wind speed in the model compared to the lidar-based profiles at $200\,m$ height during these periods. Apart from these nocturnal stably-stratified shear layers with

high wind speeds aloft, which are known to be challenging for the mixing parameterizations of mesoscale models, the overall evolution of the modeled wind profiles matches well the observed ones. The passage of the aforementioned trough and the accompanied shift in wind direction are accurately represented. The influence of low pressure resulted in unsettled weather conditions throughout the simulation time. Based on the detection of clouds in the profiles of attenuated backscatter from lidar observations at the TROPOS site (Fig. 9), a variable cloud cover prevailed during the first simulation day of 1 March, while on

2 March a significant sunshine period occurred around midday, before high-level cloud cover increased in the afternoon. Also the mesoscale simulation shows a similar evolution of cloud fraction (gray line in Fig. 10). Based on additional ground-based observations, small amounts of rain fell on 1 March during 12:00 UTC (which is missing in the model), and during the night from 1 to 2 March, with measured and modeled totals reaching nearly $4\,mm$ and $3\,mm$, respectively, at the end of the simulation period (Fig. 10). This amount is not considered to have a larger impact on the local air quality. The vertical $\Theta_v$ distribution

(area plot in Fig. 11b) indicates different influences on the PBL stratification. Intermittent cloudiness during nighttime allowed for limited surface radiative cooling, while during the sunny periods surface heating caused the $\Theta_v$ gradient to diminish as the result of convective conditions. Striking is the warm-air advection just before the trough axes crossed the area (around 2 March, 00:00 UTC, and after the end of the simulation), which resulted in a large $\Theta_v$-gradient within the lowermost $300\,m$. The simulated PBL stratification from the coarsest mesoscale domain M1 was evaluated with in-situ measurements from radio

sondes released at two meteorological sites in central Germany (also depicted in Fig. 11b). The comparison shows a generally good agreement of the model with observations, except for two cases. On 2 Match at 06:00 UTC, a too early surface heating in the model is indicated, and on 2 March at 18:00 UTC, a quite sharp inversion layer from cloud-top cooling was present at roughly $1\,km$, but missing in the model. Both discrepancies are a result of inaccurately represented low-level clouds in the coarse model run ($2.2\,km$ horizontal grid spacing), which indeed became evident from comparing the vertical profiles of

relative humidity (not shown).

The combined influences of vertical wind shear and stratification caused diverse turbulent PBL conditions and considerable height variations of the mixed layer. In Fig. 11c, the modeled bulk-Richardson number $R_i$ is shown, on which basis the MLH was computed (Eq. 2 in Vogelezang and Holtslag, 1991, with $zs = 60\,m$). For the convective period on 2 March, the additionally used parcel method gives a slightly higher MLH compared to the bulk-Richardson approach, which results in a

better agreement with the MLH derived from observations with the lidar instruments at the TROPOS site. Therefore, the red dashed line in Fig. 11c shows the maximum model-based MLH of both methods. There is a remarkable agreement with the aforementioned remote-sensing data, when considering the significant uncertainties of the retrieval methods applied on both

the model and remote-sensing data. During the early morning hours of 1 March, strong wind shear combined with an only weakly stable stratification resulted in a turbulent PBL with an estimated MLH of about $600\,\mathrm{m}$. Conditions were also quite similar on 2 March from 00:00 UTC to 06:00 UTC. Periods of warm-air advection significantly lowered the MLH even further on both simulation days after 18:00 UTC. In contrast, periods with significant solar irradiation resulted in convective conditions ($R_i < 0$), which, combined with the wind shear, increased the MLH to about $1.5\,\mathrm{km}$ on 1 March around 12:00 UTC. The PBL was even more convective on 2 March around 12:00 UTC, but due to the very limited wind shear and a more stable air mass aloft, the MLH only reached about $1\,\mathrm{km}$. While in the mesoscale simulation M3, PBL turbulence is generally unresolved, the most energetic eddies can be resolved outside the urban canopy with the $40\,\mathrm{m}$ grid spacing of domain L0. This is indicated by the vertical distribution of grid-scale and subgrid-scale TKE in Figure 12. However, from 2 March, 18:00 UTC to 23:00 UTC, and near the end of the simulation period, PBL turbulence was very weak or absent in the model as a result of the stable stratification.

In Figure 13, qualitative model results for two contrasting PBL states are featured as simulated with the CAIRDIO domain L0. In the first case (Fig. 13a), the dominant horizontal wind component $v$ near the surface is shown for the stably-stratified, shear-driven PBL on 2 March at 00:30 UTC. Turbulence is organized into horizontal streaks near the surface as it is typical for a shear-driven PBL. The effect of significant surface roughness from the city structure and forest patches locally reduced the wind speed. Over these areas, turbulence was also of more intermittent nature due to the reduced vertical wind shear near the surface and limited turbulent energy production. The largest buildings, e.g., from factories can already be resolved with $40\,\mathrm{m}$ grid spacing, while the model representation of the air flow through the diffuse city structure consisting of smaller building units is more typical of a porous-media flow (see corresponding insets). Clearly, building-induced turbulence within the diffuse urban canopy cannot be resolved using such a still comparatively coarse grid spacing, but its mixing effects are represented by diffusion from the subgrid-scale model. The second case (Fig. 13b) features a convective PBL during midday of 2 March, which is induced by the positive surface-heat flux in the model. As a result of the calm wind conditions, convection organized into open cellular structures as depicted by vertical wind speed. These structures were more prominent over the extensive crop lands surrounding the city than within the city, where they were disrupted by the effects of buildings. Striking is also the absence of convection over the still seasonally cool lakes. The contrasting PBL properties between these two cases manifest themselves in the transport of locally emitted air pollution near the surface, as shown by the horizontal maps of BC concentrations in Fig. 13(c-d). In the case of the stably-stratified PBL, transport and mixing is mostly horizontal and as a result, locally concentrated sources, like from industry, generate long down-wind tails of elevated BC concentrations. Also modeled BC background concentrations in the stable case are relatively high ($0.3\,\mu\mathrm{g\,m^{-3}}$) compared to the convective case ($0.1\,\mu\mathrm{g\,m^{-3}}$), when the pollution is effectively diluted within a much deeper mixed layer. In the convective case, the rapid vertical mixing also limits the extend of horizontal dispersion from local sources. The resulting sharp horizontal gradients lead to a clearly visible imprint of the traffic network in the horizontal map of BC concentration, which dominated the emissions during this time. To support this qualitative discussion, the domain-averaged vertical turbulent flux of scalar BC is shown for both cases in Figure 14. In the stably-stratified case, vertical mixing is limited near the surface and only gradually increases with height. In contrast, the vertical flux in the convective case already peaks close to the surface and then gradually decreases with increased height,

indicating an efficient lifting of the near-surface air pollution. Also in the convective case, the flux is much larger compared to the stable case, partly also due to the higher traffic emissions during daytime. Lastly, the partitioning into the resolved and parameterized fluxes shows that while the resolved flux is always dominant outside the urban canopy (roughly the first $30\,\mathrm{m}$), the subgrid-scale flux has also a significant contribution, mainly close to the surface and in the stable case. This also indicates a significant model sensitivity to the mixing parameterization (e.g., the prescription of the static mixing length).

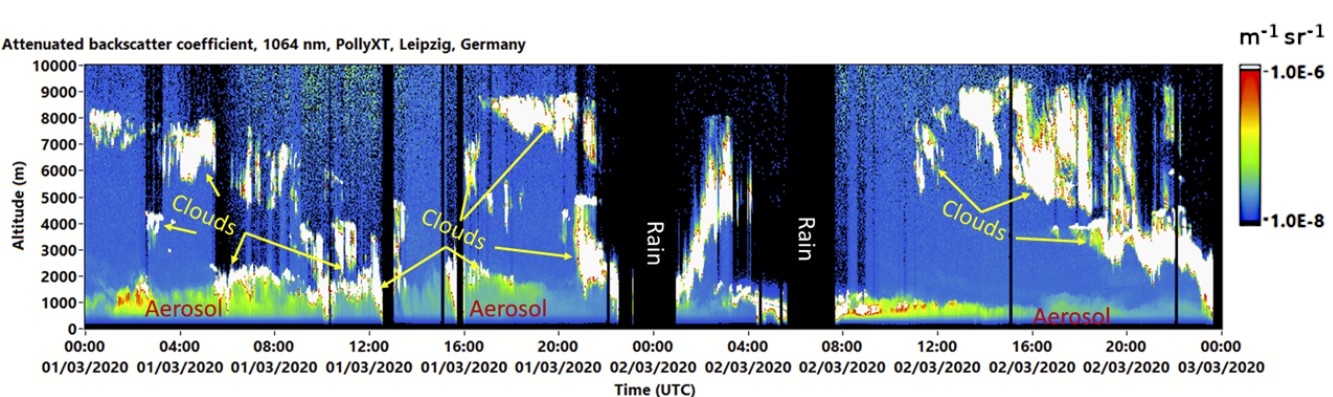

**Figure 9.** Detection of clouds, aerosol layers and periods with precipitation based on the vertical profiles of attenuated backscatter from the Polly-XT lidar at the TROPOS site.

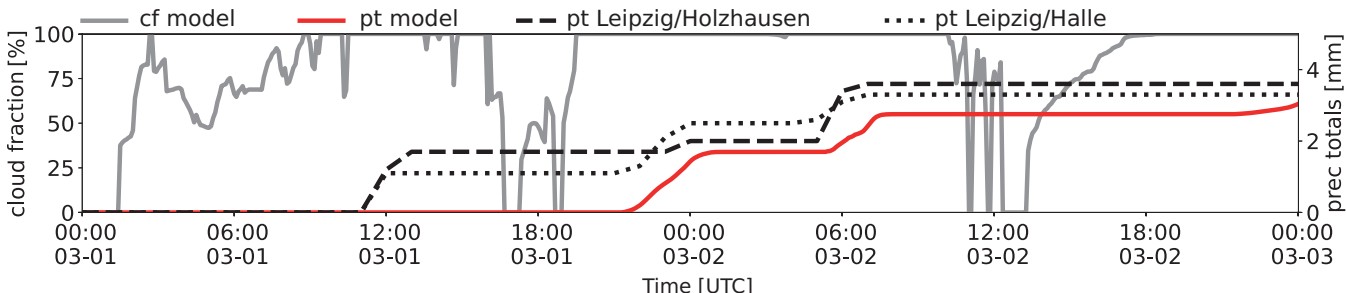

**Figure 10.** Modeled cloud fraction (cf, full grey line) and cumulative precipitation totals (pt, full red line) for Leipzig based on simulation M3. In addition, cumulative precipitation totals are shown for the meteorological sites Leipzig/Holzhausen (51.3151 °N, 12.4462 °E, dashed line) and Leipzig/Halle (51.4348 °N, 12.2396 °E, dotted line).

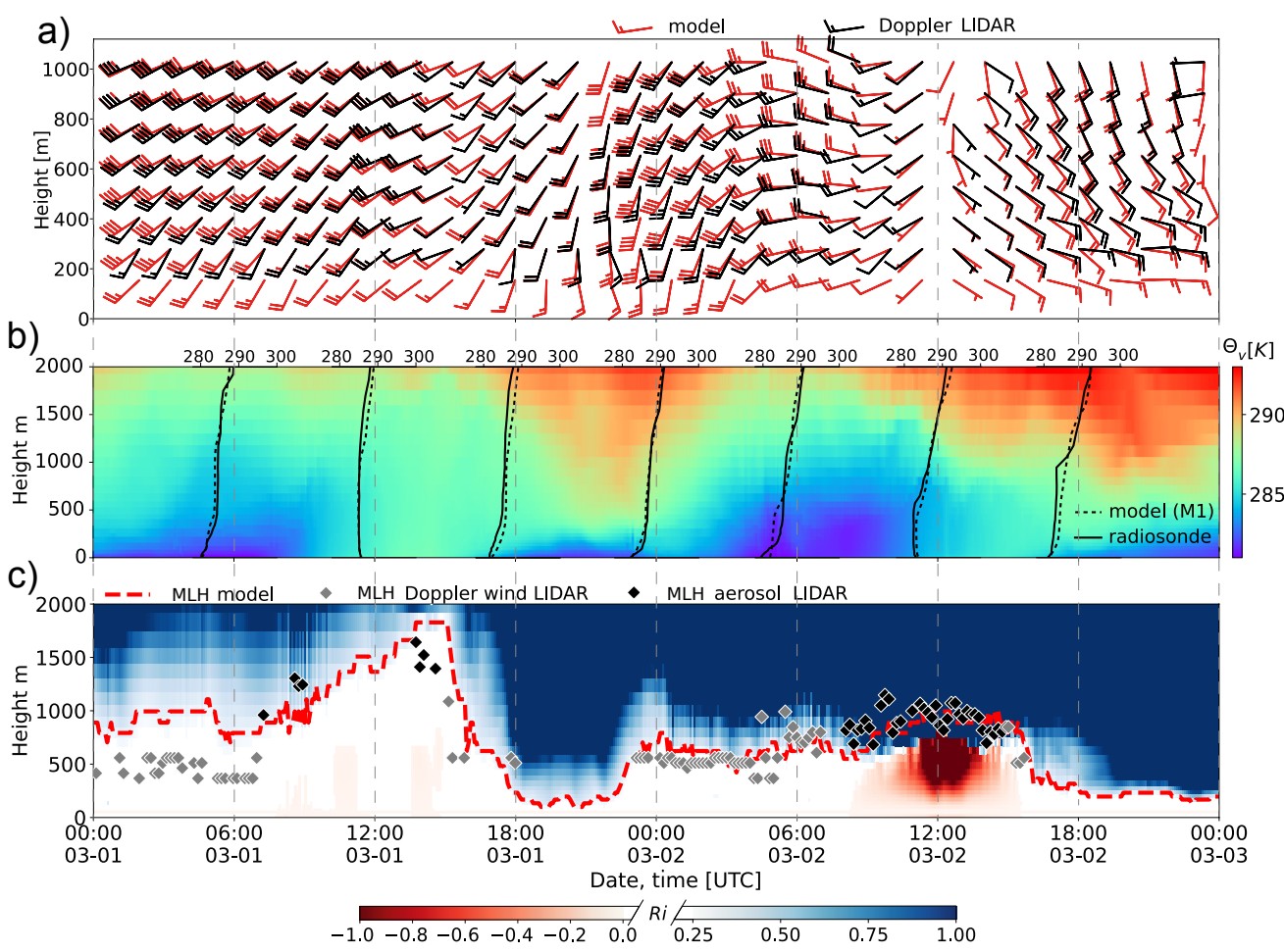

**Figure 11.** Overview of the PBL structure during the simulation period based on model and observational data (listed in Sect. 2.2). Unless mentioned otherwise, data is representative for the TROPOS site and model data are from simulation M3. In (a), vertical profiles of the horizontal wind derived from measurements with the Doppler wind lidar (black barbs) are compared to interpolated modeled profiles (red barbs). The temporal averaging period of the depicted data is 20 min. In (b) the vertical distribution of modeled virtual potential temperature $\Theta_v$ is shown by the area plot. It is supplemented with 6-hourly $\Theta_v$-profiles from atmospheric soundings at the meteorological sites Lindenberg and Meiningen (station average, full black lines), and comparable model output from domain M1 (black dotted lines). (c) shows the simulated MLH (red dashed line), which is based on a combination of the bulk-Richardson number $R_i$ (area plot) and the parcel method. Respective MLH retrievals from the wind lidar are shown by the gray diamonds, while the black diamonds show the MLH retrievals based on data from the aerosol lidar Polly XT.

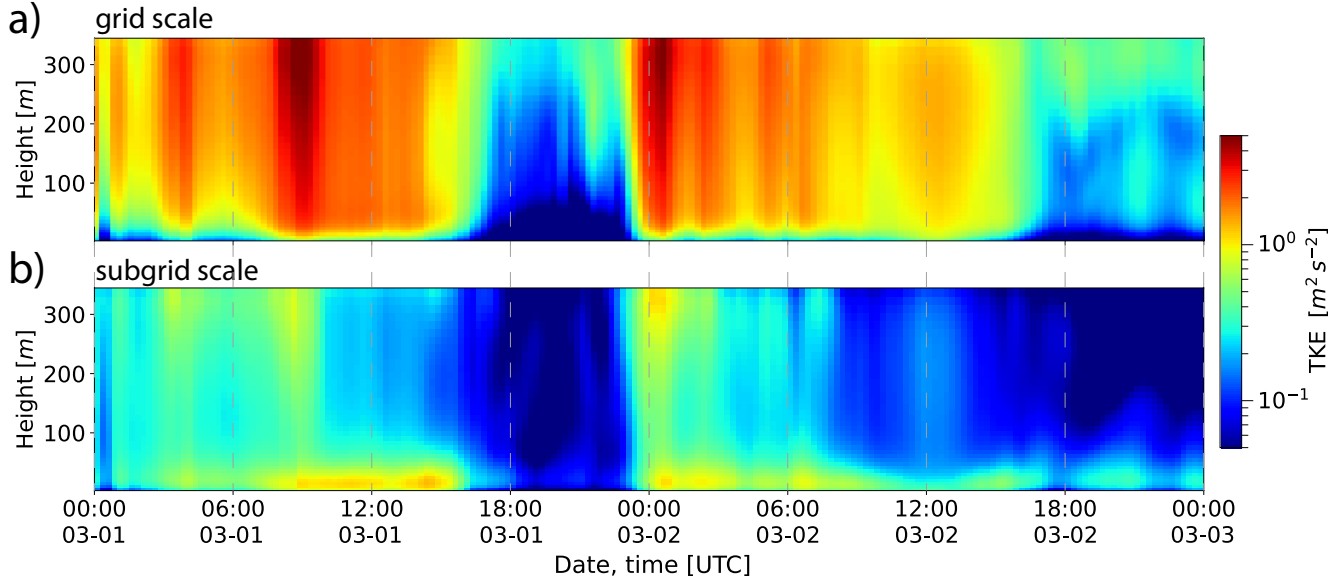

**Figure 12.** Horizontally averaged vertical distribution of grid-scale and sgs TKE versus time for simulation L0.

## 3.2 Quantitative model comparison with air-monitoring measurements

To quantitatively evaluate the model representation of the intra-urban variability of air pollution, hourly averaged model output
of $PM_{10}$ and BC concentrations are compared to respective measurements at the different air-monitoring sites within the city
area of Leipzig. For an evaluation of the mean urban wind field and air temperature, according data from two urban air-
monitoring sites is additionally compared with model data in Appendix B. Model output from mesoscale simulation M3 with
$550\,m$ horizontal resolution is added to the comparison to better quantify the benefit of the dynamic downscaling with $40\,m$
horizontal grid spacing and explicit building representation with the CAIRDIO L0 domain.

In Figure 15, respective plots of $PM_{10}$ and BC concentrations are shown for all monitoring sites in Leipzig (see details in
Sect. 2.2). For the background station LT (Fig. 15a), the measured BC profile shows a clear diurnal cycle, with the lowest
concentrations of about $0.1\,\mu g\,m^{-3}$ during the morning hours of 1 March and 2 March around 06:00 UTC. Concentrations
consistently peaked around 18:00 UTC on both days, which can be explained by the coincidence of high traffic emissions
related to the rush-hour (see e.g., prescribed emission profile in Fig. 6) and a more shallow stably-stratified PBL during this
time (see again Fig. 11e). In this respect, the morning peak on 2 March was damped by the increased PBL mixing height.
On the morning of 1 March, which was a Sunday, no peak occurred at all due to the negligible traffic emissions at this time.
Both the profiles from the mesoscale CTM and CAIRDIO simulations followed this observed evolution remarkably well. In
the temporal mean, however, the COSMO-MUSCAT simulation tend to underestimate background BC concentrations based
on the fractional bias (FB) $= 0.28$, which is improved in the CAIRDIO simulation to FB $= 0.07$. Nevertheless, the reasonably

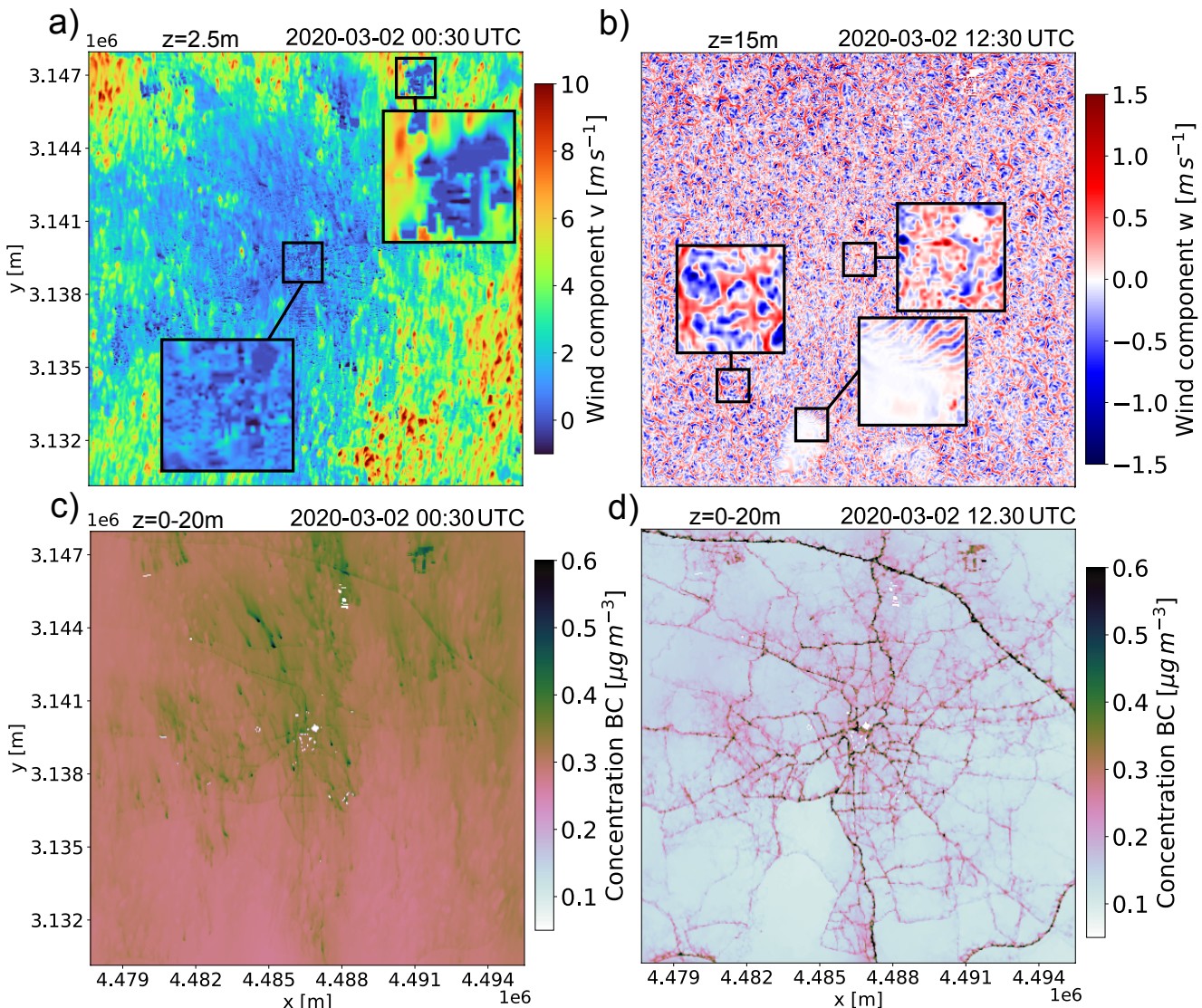

**Figure 13.** Horizontal map plots of simulation results with domain L0 (40 m) for two contrasting PBL cases: (a) shows the near-surface wind component $v$ for a stably-stratified PBL, (b) the vertical component $w$ for a convective PBL. The insets show a local magnification of some interesting flow features. In (c) and (d), the corresponding concentration fields for BC averaged within the height range 0 m-20 m are shown.

accurate model results can be considered to be a good basis for the following discussion of the roadside stations LE and LC shown in Fig. 15b-c.

Compared to the background profile, the diurnal peaks are much more pronounced at the street-canyon site LE, with peak concentrations reaching up to $2.3 \, \mu g \, m^{-3}$ at 18:00 UTC on both days. The morning peak on 2 March is again much lower

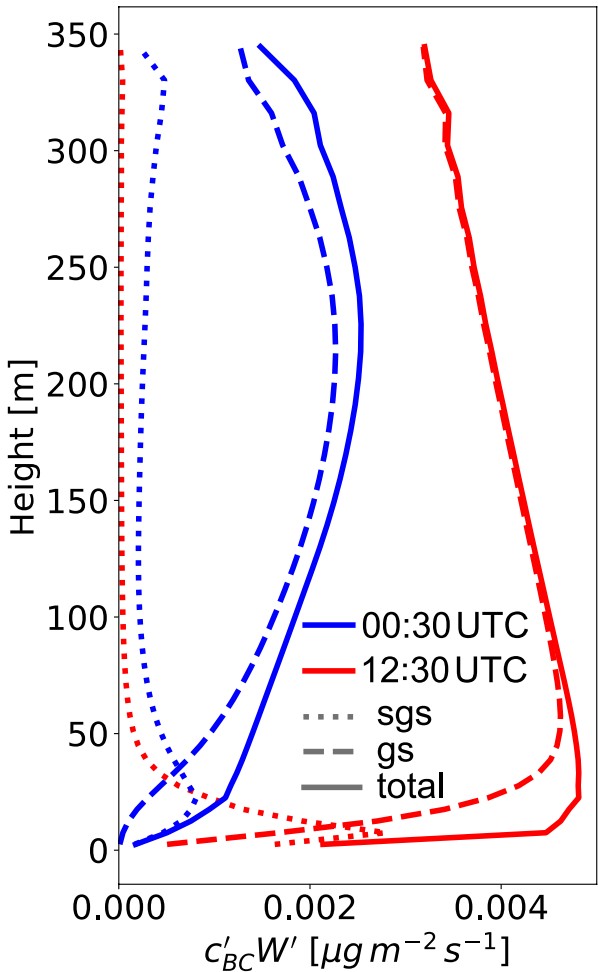

**Figure 14.** Vertical mixing of scalar BC within the PBL as simulated with CAIRDIO L0 on 2 March at 00:30 UTC featuring a stably-stratified PBL (blue lines) and on 2 March at 12:30 UTC featuring a convective PBL (red lines). Dashed lines show grid-scale mixing, dotted lines the sgs contribution, and full lines total mixing consisting of grid-scale plus sgs contribution.

compared to the evening peaks, and night-time concentrations largely correspond to the observed urban background. For the
site LC, which is situated in a more open environment, peak concentrations are expectedly lower compared to the site LE, or not
pronounced at all. This can be most likely explained by the variable influence of the nearby traffic emissions depending on the
prevailing wind direction. For example, on the second simulated evening, easterly winds advected cleaner air from the adjoining
park to the site (Fig. B1c). At both sites, the mesoscale simulation largely fails to capture the concentration peaks related to the
nearby traffic emissions, as it essentially reproduces the same background profile depicted in Fig. 15a. As a consequence, a large
positive bias results at both stations (FB $= 0.70-0.90$). In contrast to the mesoscale simulation, the CAIRDIO $40\,\mathrm{m}$ simulation

shows a much more realistic profile at the site LE, which is clearly distinct from the background profile. While the observed evening peaks are still underrepresented, the morning peak and the elevated concentrations throughout the day of March 2 are modeled remarkably accurate. Also the gradual decline of concentrations during the night hours of early 2 March follows the observed profile very well. During the morning hours of 1 March, however, modeled concentrations are too high, which can be most likely attributed to the too high prescribed emissions. As a result of the stated improvements, the bias to measurements is reduced to FB $= 0.24$ in the CAIRDIO simulation at this site. For the site LC, the CAIRDIO simulation seems to better catch up to the measured peak concentrations, which is especially the case for the first observed evening peak. On the other hand, there are also concentration peaks apparent in the simulation that were not observed (e.g. morning peak on 1 March, evening peak on 2 March). The false peaks result in a moderately negative bias of the CAIRDIO model time series in reference to the measurements (FB $= -0.25$). While the discussed pollutant BC largely behaves like a passive scalar only subjected to physical deposition, the subsequent analysis of pollutant $PM_{10}$ incorporates a much larger pool of model uncertainties related to the more diverse sources and also complex precursor chemistry of secondary aerosol. In this regard, it is not surprising to observe an already larger model uncertainty for the background profile at the site LW (Fig. 15d). While the measured profile shows only a small diurnal variability with significant noise superimposed on, the modeled profiles show more qualitative similarities with the modeled BC profiles as with the observations (i.e. smooth profiles with a significant diurnal cycle consisting of flat peaks around 18:00 UTC on both days). The reason for the missing observed short-term noise in the model results may be from unknown local sources not represented in the emission datasets used. Model biases of both the mesoscale and CAIRDIO simulations are negative (-0.32 and -0.38, respectively), indicative of an overestimation of $PM_{10}$ concentrations in the temporal mean. The measured $PM_{10}$ time series at the street-canyon site LL (Fig. 15e) exhibits a significant diurnal variability. Again, the peaks at 18:00 UTC are suggestive of a significant influence of nearby traffic emissions. In fact, the profile shares, apart from the aforementioned noise, many characteristics with the measured BC profile at site LE. Not unexpectedly, the mesoscale simulation shows again little difference to the modeled background profile, which results in a significant positive model bias (FB $= 0.50$) at this site. A large improvement can be again observed when switching to the CAIRDIO simulation, which captured the diurnal variability of roadside $PM_{10}$ concentrations very well. Only the maximum peak values during the first evening are underestimated. Correspondingly, the model bias is only slightly positive (FB $= 0.14$). Finally, the measured $PM_{10}$ profile at site LC (Fig. 15f) is more comparable to the measured background at site LW, which is a bit surprising given that the station is classified as roadside. In anyway, measured $PM_{10}$ concentrations seem to be more influenced by other not well-known sources than road traffic, at least for the investigated time period. As a result both models have their difficulties in representing the observed profile. The mesoscale simulation this time is in overall better agreement with the measured profile (FB $= 0.05$) compared to the CAIRDIO simulation (FB $= -0.40$), but likely only as a result of the overestimated $PM_{10}$ background, which by chance matches the measured roadside profile in the temporal mean.

Concluding from this analysis, BC background concentrations are represented reasonably accurate in both the mesoscale CTM COSMO-MUSCAT and the urban-scale model CAIRDIO throughout the simulation period. In contrast, the more complex pollutant $PM_{10}$ showed higher uncertainties and a considerable negative bias, and proofed therefore to be more complicated in the model study of the intra-urban air-pollution variability. As expected, the mesoscale model indiscriminately reproduced

the background profiles at all sites, which results in a large model bias for the roadside stations, except for PM$_{10}$ at the side LC. In comparison, the CAIRDIO simulation considered the influence of the local environment, as simulated roadside profiles show a much larger variability than the corresponding background profile. The model bias is within a moderately positive to moderately negative range at the roadside stations (except again for PM$_{10}$ at side LC). For BC, this residual bias can be mostly attributed to individual misrepresented peaks in the simulation. Whether a horizontal grid spacing finer than $40\,\mathrm{m}$ can still improve model representation of roadside concentrations is explored within the framework of the subsequent sensitivity study.

## 3.3 Grid-size sensitivity

### 3.3.1 Planetary boundary layer and mixing

Before evaluating grid-size sensitivity of modeled air pollution at the air-monitoring sites, some of the most important variables characteristic to the PBL state are investigated in this paragraph. For this purpose, domain LE is simulated with horizontal grid spacings of $40\,\mathrm{m}$, $20\,\mathrm{m}$, $10\,\mathrm{m}$, and $5\,\mathrm{m}$. Note that due to imperfections with the offline-nesting, the $40\,\mathrm{m}$ grid spacing is repeated with the local domains for a more accurate comparison. Resolved fluxes of momentum and tracer concentration BC are computed by using corrected temporal samples to represent an ensemble averaging. Given a scalar variable $c$, each temporal snapshot $c_i$ is corrected for horizontally-averaged changes throughout the averaging period consisting of $n$ snapshots:

$$c_i^* = c_i - <c_i>_h + <c_0>_h \; \forall i \in \{0, 1, 2, .., n\}, \tag{4}$$

where the brackets $<>_h$ denote for the horizontal averaging and $c_i^*$ is the corrected snapshot. Note that the horizontal average within the canopy layer excludes the inaccessible grid-cell volume by using the volume-scaling field $\chi$ as weights. For the velocity components, the cell-face area scaling field $\boldsymbol{\eta}$ is used instead of $\chi$. Vertical profiles of the computed horizontally and hourly averaged variables $\overline{u}$, $\overline{v}$, $\overline{\Theta_v}$, $\overline{c_{\mathrm{BC}}}$, $\overline{u'w'}$, $\overline{v'w'}$, $\overline{\Theta'w'}$, and $\overline{c'_{\mathrm{BC}}w'}$ are depicted in Figure 16 for the two contrasting PBL states already discussed in Section 3.1. Starting with the first case on 2 March at 00:30 UTC, strong southerly winds along with a weakly stable stratification created a shear-driven, turbulent PBL. In the profiles of the mean horizontal wind components (Fig. 16a-b), grid sensitivity is mainly restricted to the first $20\,\mathrm{m}$ within the canopy layer. Inside there, the run with default grid spacing of $40\,\mathrm{m}$ results in a slightly higher wind speed compared to the runs with a better resolution of buildings. Profiles of $\Theta_v$ show negligible sensitivity (Fig. 16c), while BC concentrations within the urban canopy are slightly higher in the $20\,\mathrm{m}$ and $10\,\mathrm{m}$ runs compared to the $40\,\mathrm{m}$ and $5\,\mathrm{m}$ runs (Fig. 16e). Significantly more sensitivity is observed in the vertical turbulent fluxes of momentum (Fig. 16f-h), virtual potential temperature (Fig. 16i) and scalar BC (Fig. 16j). Although the subgrid-scale contributions (dotted lines) become larger as the resolution is decreased, they do not seem to compensate for the loss of resolved fluxes. Apparently, this issue is not restricted to the urban canopy, but may be influenced by an underestimation of vertical wind shear just above the roof tops in the coarser runs. Nevertheless, sensitivity in $c_{BC}$ and $\Theta_v$ is very low, arguably because transport is mostly horizontal in the shear-driven case. Thus, the profiles of $c_{BC}$ and $\overline{c'_{\mathrm{BC}}w'}$ (respective $\Theta_v$ and $\overline{\Theta'_v w'}$) are only weakly related to each other.

For the featured convective case around 2 March 12:30 UTC, horizontal wind speeds are by one order of magnitude lower compared to the first case, but a positive vertical heat flux (Fig. 16s) is responsible for turbulence generation this time. Notably, the heat flux sharply increases from the surface up to the average roof height, which highlights the heating effects from building walls and roofs. The averaged wind profiles (Fig. 16k-l) show a similar sensitivity to the already discussed shear-driven case, which may indeed indicate an increase of the prescribed roughness length of the subgrid-scale building structure for momentum transfer in future simulations with diffuse buildings. Compared to the shear-driven case, a more substantial sensitivity in $\overline{\Theta_v}$ of approximately $0.5\,\mathrm{K}$ and also in $\overline{c_{BC}}$ can be observed for the height range within the urban canopy (Fig. 16m, o). In contrast, a negligible grid-sensitivity is observed for the respective vertical fluxes $\overline{\Theta_v' w'}$ and $\overline{c_{BC}' w'}$ (Fig. 16s-t) across the full height range. The reason for this behavior in the given convective case is that scalar transport is dominated by vertical mixing. Since the emissions are constant in all simulations, also no variations in the turbulent flux $\overline{c_{BC}' w'}$ are expected when assuming an equilibrium state. This, however, does not imply that the profiles of eddy diffusivity are constant across the simulations, as the vertical gradients in $\overline{c_{BC}}$ adjust to match the flux profile. In fact, there must be a significant sensitivity of the vertical eddy diffusivity within the height range where the scalar profiles start to diverge, which is not just by coincidence also the area with the largest instability (the largest super-adiabacity in $\overline{\Theta_v}$). In the currently used subgrid-scale model, the turbulent Prandtl number $P_t$, which relates the eddy diffusivity to the eddy viscosity, is not further decreased below the neutral value of 0.66 for unstable stratification. However, in unstable conditions $P_t$ may be actually much lower, implying a larger eddy diffusivity. An adjustment of the stability-dependent $P_t$ may be needed in future simulations.

### 3.3.2 Distribution of air pollutants

For the evaluation of grid sensitivity at the air-monitoring sites, the horizontal grid spacing of the locally nested domains centered at roadside-classified air-monitoring sites (LE, LL, or LC, respectively) is varied between $40\,\mathrm{m}$, $20\,\mathrm{m}$, $10\,\mathrm{m}$, and $5\,\mathrm{m}$. The finest grid spacing of $5\,\mathrm{m}$ permits conventional building-resolved simulations, but is omitted for the background stations LT and LW due to the expected low model sensitivity there. Model results are again hourly averaged and spatially interpolated to the exact locations of the measurement sites. In Figure 18, the obtained time series are plotted against each other and against the measurements as reference, similar to Fig. 15. In addition, the corresponding FB values in relation to the measurements are listed in Tab. 3. For the BC background station LT (Fig. 18a), little grid-size sensitivity is found as to expect. At best, BC peak concentrations around 1 March 18:00 UTC tend to be slightly higher with increased resolution, which is mainly from the sensitivity of the subgrid-scale parameterization, as the PBL was stably stratified during this time. FB varies only slightly from 0.05 to 0.00 for the $10\,\mathrm{m}$ grid spacing. More interesting are the results for the roadside station LE (Fig. 18b). For this site, the $40\,\mathrm{m}$ grid spacing resulted in an underestimation of BC peak concentrations. In fact, decreasing the grid spacing down to lower or equal $10\,\mathrm{m}$ results in a significantly better representation of both evening rush-hour peaks. However, absolute peak concentrations still cannot be fully recovered. Interestingly, no further improvements can be achieved with the finest grid spacing of $5\,\mathrm{m}$, which points towards limitations with the used emissions. Nevertheless, the improved peak representation results in a slightly lower model FB ranging from 0.06 to 0.16. Aside from the discussed rush-hour peaks, model sensitivity is considerably lower, and it seems like the $10\,\mathrm{m}$ and $20\,\mathrm{m}$ grid spacings result in slightly higher concentrations compared to the

40 m and 5 m grid spacings, which are close together. Most likely, this particular sensitivity, which is also observed at other sites (e.g. LC in Fig. 18c), can be attributed to the static subgrid-scale mixing length $\Delta_{sgs}$. In the 40 m run, $\Delta_{sgs}$ is 20 m near the ground, which is about the typical height of buildings, and thus adequate when considering that the eddies within the urban canopy cannot be resolved. In the 20 m run, however, $\Delta_{sgs}$ is significantly smaller than building size, while resolution is still too coarse to capture the most important eddies. This likely results in an underestimation of vertical mixing within the urban canopy at such intermediate resolutions. By further decreasing the grid spacing down to 10 m or below, the largest eddies of the turbulent canopy flow are finally resolved.

For the station LC (Fig. 18c), a distinction of the evening rush-hour peaks from the remaining time series turns out to be reasonable too. For the two evening peaks, a decrease of peak BC concentrations with increasing spatial resolution can be observed, which is in contrast to the sensitivity at the station LE. A closer inspection of the spatial concentration gradients in Figure 17 reveals the reason for this contrasting sensitivity. The station LC lies next to two traffic lanes to the north in the model. In the 40 m run, the exhaust plumes spread over a comparatively large area, causing a smearing of the gradient in the vicinity of the air-monitoring site. In the 5 m run, spatial gradients near the road are much sharper, which places the measurement site mostly outside the exhaust plumes. It is, however, questionable if the line-source representation of traffic emissions is still adequate in combination with such a fine grid spacing, as in reality emissions can be effectively spread over a larger area by car-induced turbulence (Gross, 2016).

At the site LE, BC is directly measured above the traffic emissions. Here, the 40 m horizontal grid spacing is too coarse to keep the air pollution trapped within the narrow street canyon, as also part of the emissions are emitted outside of the canyon. This explains the observed positive sensitivity of modeled peak BC concentration with increased resolution. Obviously, the 10 m grid spacing is already adequate to contain all of the traffic emissions within the canyon. A similar observation is made with respect to the wind direction at street-canyon site LL (see Fig. B1d in Appendix B).

Grid-sensitivity at the $PM_{10}$-measurement site LW is negligible (Fig. 18d), while it is again much more significant at the street-canyon site LL, where bulk $PM_{10}$ is influenced to a large degree by traffic emissions. Not surprisingly, decreasing the grid spacing to 20 m results in higher modeled peak concentrations compared to the 40 m grid spacing. A further decrease of the grid spacing leads to some indecisive changes for the first evening peak and no significant changes for the second peak. As a result of the higher modeled peak concentrations in the higher resolved runs, an initially moderately positive FB $= 0.21$ of the 40 m is turned into slightly negative FB values ranging from -0.11 to -0.19. Finally at site LC, a quite similar behavior to the already discussed pollutant BC can be observed, albeit with an extenuated amplitude from the higher background influence of $PM_{10}$. Hence, FB varies only slightly between -0.46 and -0.32 for the set of sensitivity runs.

Having discussed in detail the grid-sensitivity of modeled BC and $PM_{10}$ at the different measurement sites, it remains to quantify this sensitivity in proportion to the simulation error in order to give a final conclusion. In addition, also the Pearson correlation coefficient $r$ can serve as a criterion to judge the agreement of model results with observations. As metrics of the

model error, both the root-mean-square error ($\epsilon_{RMS}$, units of $\mu\mathrm{g\,m^{-3}}$) and the relative error ($\epsilon_r$, units in $\%$) are computed by the following equations:

$$\epsilon_{RMS} = \sqrt{\frac{\sum_t \{c_{mod} - c_{obs}\}_t^2}{n}} \tag{5}$$

and

$$\epsilon_r = \frac{\sum_t \{|c_{mod} - c_{obs}|\}_t}{\sum_t \{c_{obs}\}_t} \times 100\,\%, \tag{6}$$

where $c_{mod}$ and $c_{obs}$ are the modeled and measured concentrations, respectively. $t$ is the time index and $n$ the number of time steps or observations. $\epsilon_{RMS}$, $\epsilon_r$ and $r$ values computed for all sensitivity runs are additionally listed in Tab. 3. Based on these additional criteria, a large part of the mismatch between observations and measurements cannot be tackled by simply increasing the spatial resolution of the simulation. The largest error reduction and increase in $r$ still results at the street-canyon sites. For example, both $\epsilon_{RMS}$ and $\epsilon_r$ of modeled BC concentrations at site LT are roughly one quarter lower in the $5\,\mathrm{m}$ simulation compared to the $40\,\mathrm{m}$ run. This suggests that grid spacing has the largest influence in close proximity to important pollution sources, like traffic. Still, the model error is dominated by other influences even there. At the other measurement sites this is even more so the case, as a relevant error reduction or increase in $r$ cannot be observed when refining the grid. Concluding from these results, the benefits of a decrease in horizontal grid spacing beyond $40\,\mathrm{m}$ seem only minor in this limited modeling study. While in principle, processes can be resolved more accurately using a fine grid spacing, the potential advantages in real simulation studies are often scotched by the error from other processes, which are currently represented in a very simplified form in the model. For example, the emission model not only crudely simplifies the composition and spatial distribution of different emission types, but also their activation based on a smooth temporal profile, which cannot respond to locally deviating conditions.

## 4 Conclusions

In this study, we applied the dispersion model CAIRDIO for the first time on a real mid-sized city to simulate dispersion of the pollutants $PM_{10}$ and BC using a realistic meteorological setup, which was interpolated from a hosting mesoscale simulation. For the simulation period, two consecutive days in early March 2020 were selected. During this time, unsettled weather conditions with changing PBL characteristics and a generally pronounced magnitude of the intra-urban variability due to relatively low background pollution concentrations prevailed. The horizontal grid spacing of the model was set uniformly to $40\,\mathrm{m}$, which permits only to resolve the largest building structures, like industrial sites, while the majority of buildings within the city is described as diffuse obstacles. Nevertheless, the LES approach allows for an explicit representation of the most important turbulent PBL processes, which also include effects from a thermal surface forcing essential to the evolution of the PBL. This capability of the dynamical approach can be considered as a major advantage over more idealized models considering such

| | LT-BC | LE-BC | LC-BC | LW-PM$_{10}$ | LL-PM$_{10}$ | LC-PM$_{10}$ |
|---|---|---|---|---|---|---|
| **FB** | | | | | | |
| 40 m | 0.05 | 0.27 | -0.19 | -0.37 | 0.21 | -0.36 |
| 20 m | 0.01 | **0.06** | -0.29 | -0.38 | **-0.11** | -0.46 |
| 10 m | **0.00** | **0.06** | -0.20 | **-0.36** | -0.19 | -0.41 |
| 5 m | - | 0.16 | **-0.05** | - | -0.14 | **-0.32** |
| **$\epsilon_r$ [%]** | | | | | | |
| 40 m | 30.5 | 40.3 | 39.2 | 52.8 | 29.2 | 61.2 |
| 20 m | **30.2** | 37.8 | 46.5 | 54.7 | 28.6 | 76.7 |
| 10 m | **30.2** | **30.4** | 41.0 | **52.6** | 31.2 | 70.8 |
| 5 m | - | 32.9 | **35.1** | - | **27.8** | **60.0** |
| **$\epsilon_{RMS}$ [µg m$^{-3}$]** | | | | | | |
| 40 m | 0.16 | 0.41 | 0.27 | **3.06** | 4.76 | 5.11 |
| 20 m | **0.15** | 0.36 | 0.31 | 3.15 | 4.51 | 6.27 |
| 10 m | **0.15** | **0.29** | 0.26 | 3.07 | 4.71 | 5.84 |
| 5 m | - | 0.31 | **0.23** | - | **4.38** | **4.94** |
| **$r$** | | | | | | |
| 40 m | **0.87** | 0.88 | **0.73** | **0.71** | 0.66 | **0.46** |
| 20 m | 0.86 | 0.84 | 0.70 | 0.69 | 0.73 | 0.19 |
| 10 m | **0.87** | **0.90** | 0.70 | 0.68 | **0.77** | 0.17 |
| 5 m | - | **0.90** | 0.68 | - | **0.77** | 0.29 |

**Table 3.** List of model to measurement FB, model error $\epsilon$, and Pearson correlation coefficient $r$ computed for all concentration time series of the performed sensitivity runs. The best-performing model resolution for each station and criterion, respectively, is highlighted in bold.

effects, like stratification, only in parametric form (e.g. Gaussian plume or street-canyon models). In fact, the modeled PBL in this study showed turbulent features, which were consistent with the expected qualitative characteristics based on thermal stratification and vertical wind shear alone, and and it was shown that the modelled MLH agreed well with reference mea-
705 surements. Periods of intermittent or absent turbulence occurred when the critical bulk Richardson number was exceeded. The different dominating dispersion pathways during a specific PBL state (horizontal advection for shear-driven PBL vs. vertical turbulent mixing for convective PBL) resulted in visible qualitative differences in modeled near-surface BC concentrations, like a significantly higher background concentration and smoothed-out gradients in the shear-driven case compared to the convective case. The quantitative evaluation of modeled pollutant concentrations at the air-monitoring sites representative to the
710 urban background showed a diurnal variability in modeled BC concentrations that was consistent with the measurements and thus provided further evidence for a realistic model representation of PBL transport processes. The model agreement with the

measurements was also better for BC than $PM_{10}$, as BC is more locally influenced, while $PM_{10}$ includes not only predominantly regional influences, but also uncertainties in the complex precursor chemistry. Ultimately, the model representation of the intra-urban variability of BC and $PM_{10}$ concentrations was evaluated using the measurements at the road sites. These were distinct from the measurements at the background sites by the significantly elevated concentrations throughout daytime and the peaks from the traffic-rush hours. Here the model responded adequately to the different environment characterized by surrounding buildings and high localized traffic emissions, as it mostly reproduced the elevated concentrations. This was in strong contrast to the driving mesoscale simulation, which indiscriminately reproduced the background concentrations at all sites. While we did not directly apportion the effects of buildings represented as physical obstacles in addition to the increased model resolution leading to these different model results, the performance of additional sensitivity runs using locally nested domains with decreased horizontal grid spacing down to $5\,m$ helped to shed some light on this aspect. While we did not observe much model sensitivity to grid spacing at the background sites, this was not the case at the roadside locations. Part of the observed sensitivity at the street-canyon sites could be explained by the simple fact that a horizontal grid spacing of $40\,m$ is not sufficient to contain the traffic emissions within the narrower street canyons. However, without the presence of buildings in the model, the emissions would have been diluted into a much larger air volume from the first place. At more open sites, the more pronounced horizontal smearing of pollution gradients with increased horizontal grid spacing resulted in an opposite sensitivity of modeled concentrations, as the air-monitoring sites are located outside the densest traffic-emission plumes in the $5\,m$ run. Arguably, the sign of the sensitivity also depends on the distance to the nearby traffic lanes here. It is disputed, however, if the results with the finest grid spacing are more realistic, as turbulent diffusion might be underestimated by neglecting small scale processes like traffic motion. It became also apparent that buildings contribute importantly to turbulent vertical mixing within the roughness sublayer. When decreasing the grid spacing, these turbulent motions are successively better resolved. Especially in the shear-driven case, we observed a significant grid-size sensitivity of the vertical turbulent scalar flux just above the building roofs. The best explanation for the underestimation of this flux in the default $40\,m$ simulation is a possible underestimation of the drag from the diffusely resolved buildings on the air flow, which would reduce vertical wind shear above the building roofs and also decrease the vertical turbulent flux in favor of the horizontal advective flux. In the convective case, grid-size sensitivity of domain-averaged near-surface BC concentrations could also be traced back to an underestimation of the parameterized vertical diffusivity, especially in the super-adiabatic height range induced from the surface-heat flux. Here, a possible mismatch of the parameterized turbulent Prandtl number was pointed out. In order to further corroborate the reasons behind the observed sensitivity in the vertical scalar fluxes, additional sensitivity runs with variations of the parameters in question (e.g. $r_0$, $P_t$) need to be carried out in future studies focusing mainly on such aspects. Also, we assumed the validity of the downscaled surface potential-temperature fields prescribed in the simulation, which affect also the heat flux from building surfaces, mainly in lack of a more physically-based alternative. Here, the further comparison with a microscale model equipped with an own radiation and surface scheme could provide confidence, which was however out of the scope of this study. Finally, in spite of the observed and discussed sensitivity, the comparison of the error in modeled concentrations at the measurement sites showed only slight improvements with a decreasing grid spacing, if any at all. For a more significant model evaluation, definitely a more prolonged simulation period needs to be investigated. Still, the findings from this study point to

the necessity of more accurately representing other non-physical components in the model, in order to benefit from a more accurate representation of model physics with building-resolving grids. Most notably to mention are the traffic emissions with their currently limited accuracy and comprehensiveness, which may be improved in future simulations with the incorporation of real-time traffic-flux data. Nevertheless, with the currently available data the showcased modeling approach performed at urban gray-zone horizontal resolutions showed to be a very promising tool for application on more targeted research questions that previously relied on mesoscale model outputs, like, e.g., urban population exposure to air pollution.

*Code availability.* The source code of CAIRDIO model version 2.0, as well as utilities for data pre-processing are accessible in release under the license GPL v3 and later at https://doi.org/10.5281/zenodo.6075354 (Weger et al., 2022).

*Data availability.* The data used in this study, which include model results and observations, are accessible at https://doi.org/10.5281/zenodo.6077050.

## Appendix A: CAIRDIO v2.0 improvements

The actual model version 2.0 used in this paper features additional improvements over the published version 1.0, which apply to several model components and are listed in the following.

### A0.1 Revised advection scheme

In CAIRDIO v.1.0, advection used linear $5^{\text{th}}$-order reconstructions with additional limiting for positive scalars. It is well known that such upwind-biased odd-order schemes result in numerical diffusion, as the leading error term is diffusive. In LES, numerical diffusion has a detrimental impact on the correct energy transfer, as excessive energy is drained from the smallest scales that feed on the larger eddies, thus affecting the entire energy cascade. The manifestation of excessive damping can be seen in the energy spectra of Figure A1. Nevertheless, some sort of numerical damping of the smallest wavelengths is necessary in order to maintain numerical stability, as these scales are flawed by large dispersion errors. Recognizing that the standard $5^{\text{th}}$-order linear upwind formulation carried out in each dimension separately is simply the addition of a high-order diffusion term to a non-diffusive $6^{\text{th}}$-order central scheme, directly leads to an opportunity to control numerical diffusion:

$$\partial_{adv}^{5\text{th}} = \partial_{adv}^{6\text{th}} + \nabla \cdot \left( \boldsymbol{u}_+ \Delta_h \nabla^{5\text{th}} \right) \tag{A1}$$

Here, $\Delta_h$ is the grid spacing, $\boldsymbol{u}_+$ the positive definite transport velocity component, and $\nabla^{5\text{th}}$ a finite difference operator using $5^{\text{th}}$-order reconstructions. The product $\boldsymbol{u}_+ \Delta_h$ is called numerical diffusion coefficient, and mainly acts in the dominant transport direction. In the revised scheme, $\boldsymbol{u}_+$ is replaced by a constant parameter $d = 0.05$. Note that this results in isotropic diffusion, similar to the method proposed in Xue (2000) for high-order damping.

## A0.2 Prognostic TKE formulation for subgrid-scale mixing

In version 1.0 an algebraic eddy-viscosity formulation was used. Therein, eddy viscosity was diagnosed from the strain-rate tensor $S$ without taking buoyancy effects into account. In order to simulate non-neutral PBLs, we implemented a prognostic subgrid-scale TKE formulation similar to Deardorff 1973 in version 2.0. This scheme not only takes buoyancy effects into account but also avoids the local-equilibrium assumption and thus may provide more accurate results with coarse grid spacings. The prognostic equation for subgrid-scale TKE is given by

$$\partial_t e = -\nabla\left(\boldsymbol{u}e\right) + 2\nabla k_h \nabla e + 2k_m|\boldsymbol{S}|^2 - k_h N_c^2 - \frac{c_\epsilon}{l_{sgs}}e^{3/2}. \tag{A2}$$

The first two terms correspond to the advective-diffusive transport, which also incorporate the pressure-correlation term parameterized by a doubling of the diffusive flux. The shear-production term is parameterized with the squared magnitude of $S$ and the buoyancy-production term results from the squared Brunt–Väisälä frequency $N_c$ multiplied with the eddy diffusivity $k_c$. Finally, the dissipation term contains a stability-dependent subgrid-mixing length, which is formulated by

$$l_{sgs} = \min\left(\Delta, 0.76\sqrt{e}/N_c\right), \tag{A3}$$

where $\Delta = (\Delta_x\Delta_y\Delta_z)^{1/3}$ is the static (grid) mixing length. The eddy viscosity coefficient $k_m$ is parameterized according to:

$$k_m = c_m\sqrt{e}\Delta \tag{A4}$$

Note that we replaced $l_{sgs}$ with $\Delta$ therein, as the original formulation resulted in a too small eddy diffusivity for stable stratifications within the roughness sublayer of diffuse urban canopies. We argue that in such a case, the mixing length is 790 always at least as large as the typical building height, and thus not really stability dependent therein. $c_m$ and $c_e$ are model constants, and can be related to the static Smagorinsky constant by $c_s = c_m^{3/4}/c_e^{1/4}$. Fixing the value of $c_e = 0.93$, the constant $c_m$ is then determined by the choice of $c_s$. Thus, $c_s$ is retained as the same model parameter used in the diagnostic scheme.

Lastly, the eddy diffusivity $k_h$ for heat and scalar transport is related to $k_m$ by the inverse turbulent Prandtl number:

$$k_h = Pr_t^{-1}k_m \tag{A5}$$

In Deardorff (1973), $Pr_t^{-1}$ is parameterized according to

$$Pr_t^{-1} = 1 + 2\frac{l_{sgs}}{\Delta}. \tag{A6}$$

However, as pointed out in paragraph 3.3.1, the neutral value of $Pr_t^{-1}$ may be too low in unstable conditions (cf. Li et al. 2015), and if a negative impact from this is further corroborated, the stability dependency will be entirely replaced by another formulation in a future model version.

## A0.3 Domain-top boundary condition for velocity and scalars

An important aspect in numerical simulations of the PBL is the treatment of the domain-top boundary condition, for which absorbing sponge layers are commonly used to dampen vertically propagating waves and to nudge fields towards their prescribed mean states. However, as a sponge layer prevents resolved turbulent fluxes at the domain top, it's use is only justified for sufficiently large vertical domain extends. In practical simulations with a variable PBL state and MLH, this requirement is difficult to maintain throughout the simulation period and would be also computationally expensive. For this reason, CAIRDIO can use an alternative domain-top boundary condition for velocity, which allows for limited vertical turbulent transport existing at the domain top. The principle of the boundary condition is similar to the turbulence recycling scheme described in Weger et al. (2021). In a first step, the turbulent velocity fluctuations within the uppermost model layer are extracted by application of a horizontal filter:

$$\boldsymbol{u}'(x,y) = \boldsymbol{u}(x,y) - <\boldsymbol{u}(x,y)>_{w_r}, \tag{A7}$$

where $<>_{w_r}$ is the filter operation with filter width $w_r$. In a second step, the turbulent fluctuations are re-scaled to match a prescribed target intensity by:

$$\boldsymbol{u}'_{tar}(x,y) = \boldsymbol{u}'(x,y) \min\left(a_{max}, \frac{||\boldsymbol{u}'_{tar}||_2}{||\boldsymbol{u}'||_2}\right). \tag{A8}$$

Here $a_{max}$ is a factor limiting the amplification of turbulence. Note that in the case of the coupling with a larger-scale parent model, $||\boldsymbol{u}'_{tar}||_2$ can be identified with $\sqrt{2E^f_{res}}$ of Section 2.4.2. Also note that if not only the subgrid-scale TKE, but instead their 3-D components are known from the PBL parameterization of the mesoscale model, a more accurate re-scaling ca be obtained (currently not used), as Eq. A8 still permits a free evolution of $u'$, $v'$, and $w'$ within the bounds of $||\boldsymbol{u}'_{tar}||_2$. The rescaled fluctuations are finally added to the average or large-scale (mesoscale) velocity components prescribed at the domain top. The nature of the described boundary condition can be interpreted as a mixed Dirichlet/Neumann boundary condition: While a pure Neumann condition for velocity does not permit any external control or forcing and is therefore also numerically unstable, in this type of mixed boundary condition the Neumann condition applies only to the small-scale fluctuating component with the additional constraint of the intensity rescaling acting as a form of numerical stabilization. The large-scale or average-state contribution is, however, imposed by a Dirichlet condition allowing for the mesoscale forcing to take effect also at the domain top. For all other prognostic scalars in the model (e.g. potential temperature, subgrid-scale TKE, tracers), a simple Dirichlet boundary condition is imposed, as the resolved turbulent motions at the domain top then act in down-mixing of the externally prescribed values.

Figure A2 demonstrates the explained turbulent boundary condition based on a sensitivity study using domain L0, and with the same mesoscale forcings and lateral boundary conditions applied as in the main L0 simulation. The averaging period for the computed vertical fluxes $\overline{w'w'}$, $\overline{c'_{BC}w'}$ (Fig. A2 a-b) and the mean concentration profile of BC $\overline{c_{BC}}$ (Fig. A2 c) refers to the period from 2 March 12:00 UTC - 13:00 UTC, when a well-defined convective PBL persisted. The simulation using a

default domain height of $350\,\mathrm{m}$ (run T350, orange lines) is compared to a control simulation with the domain height increased to $700\,\mathrm{m}$ (run T700, red lines), and a third simulation with $350\,\mathrm{m}$ domain height but using a Rayleigh damping layer instead of the turbulent boundary condition at the domain top (run D350, blue lines). While the vertical fluxes $\overline{w'w'}$ and $\overline{c'_{BC}w'}$ are somewhat smaller in run T350 compared to run T700 near the domain top, which is due to the uncertainties in the prescribed turbulent intensities, the sensitivity of near-surface BC concentrations is still negligible (less than $1\,\%$). In simulation D350, vertical fluxes at the domain top are entirely suppressed, which results in an increase of near-surface BC concentrations of about $4\,\%$ compared to run T350. Concluding from this comparison, there is a clear advantage of the turbulent boundary condition over a Rayleigh damping layer for limited domain heights.

### A0.4    Dry and wet deposition of particulate matter

For a consistent model description of particulate matter dispersion, dry and wet deposition processes are considered too in the new model version 2.0. For dry deposition, the scheme of Zhang and He (2014) was implemented, which considers the bulk-size categories $PM_{2.5}$, $PM_{2.5-10}$ and $PM_{10+}$. Accordingly, the deposition flux of a given particle category on a horizontal surface is given by the particle mass concentration $c_{pm}$ times the parameterized deposition velocity $v_d$, which contains the contributions from gravitational settling $v_g$, aerodynamic resistance $r_a$ and surface resistance $r_s$:

$$v_d = v_g + \frac{1}{r_a + r_s} \tag{A9}$$

$v_g$ primarily depends on particle size and can thus be set to a constant. For $PM_{2.5-10}$ a value of $10^{-4}\,\mathrm{m\,s^{-1}}$ is used, while for $PM_{2.5}$ gravitational settling is neglected. The aerodynamic resistance is computed according to

$$r_a = \frac{2}{3u_* t_h}, \tag{A10}$$

where $u_*$ is the friction velocity and $t_h$ the surface-transfer coefficient for heat, also used in the parameterization of heat and moisture fluxes. Finally, Zhang and He (2014) provide empirical relationships for the surface-depostion velocity $v_{ds}$, which is the inverse of $r_s$. For $PM_{2.5}$, a linear dependency on $u_*$ is assumed

$$v_{ds} = a_1 u_*, \tag{A11}$$

while $PM_{2.5-10}$ uses a cubic formula for land-use classes with constant leaf-area index:

$$v_{ds} = b_1 u_* + b_2 u_*^2 + b_3 u_*^3. \tag{A12}$$

The coefficients $a_1$, $b_1$, $b_2$, and $b_3$ are adjusted to the different land-use classes considered by the scheme. To obtain the final deposition tendency, the computed deposition fluxes for each relevant land-use class are multiplied with respective surface area

and summed up.

Since clouds are not computed in CAIRDIO, wet deposition is solely based on precipitation (sub-cloud) scavenging, for which precipitation rates $pr$ $[\mathrm{kg\,m^{-2}\,s^{-1}}]$ of the mesoscale host simulation are inferred from. It is assumed that throughout the simulated vertical range of the PBL, the precipitation rate is constant. As for the scavenging coefficients, $0.104\,\mathrm{m^2\,kg^{-1}}$ and $0.418\,\mathrm{m^2\,kg^{-1}}$ are assumed for the categories $\mathrm{PM_{2.5}}$ and $\mathrm{PM_{2.5-10}}$, respectively.

## Appendix B: Evaluation of modeled urban meteorology with near-surface observations

For the evaluation of urban meteorology in the model, near-surface observations of hourly-averaged wind speed, wind direction and air temperature are additionally available to the air-quality measurements at the air-monitoring sites LL and LC, respectively. In Figure B1 respective observations are compared with outputs from the mesoscale model run COSMO M3, the default CAIRDIO simulation L0, and the concerning nested CAIRDIO simulation with $5\,\mathrm{m}$ grid spacing at the sites LC and LL. The nested model runs allow to discuss some aspects of the sensitivity to grid spacing. Wind speed and direction at site LC (Fig. B1a, c), which is surrounded by more open areas, shows a much more complex evolution over time than suggested by the lidar observations aloft (cf. Fig. 11a). The measured wind speed is the highest (up to $5\,\mathrm{m\,s^{-1}}$) during the periods of a convectively enhanced PBL around midday of both days, and generally lower during a nocturnal shallow PBL, which can be both explained by different rates of turbulent vertical momentum transport. The most frequent wind directions are either from the west or the east with quite abrupt turnings, which indicates a significant influence of nearby buildings. Since mesoscale simulation M3 contains only parameterized building effects in combination with a $550\,\mathrm{m}$ horizontal grid spacing, it is not surprising that the modeled profiles (blue lines) do not follow the observed profiles of wind speed and wind direction very well. Significantly more realistic are, however, the according temporal profiles of the CAIRDIO simulation L0 (orange lines), as they follow the observed trend quite well, even though some underestimation of wind speed (up to $2\,\mathrm{m\,s^{-1}}$) is evident during the periods with the highest observed wind speeds. The locally nested simulation with $5\,\mathrm{m}$ grid spacing (red lines) shows little improvements at this site. For the street-canyon site LL (Fig. B1b), measured wind speeds are generally mostly below $2\,\mathrm{m\,s^{-1}}$, while the wind direction (Fig. B1d) shows a jump profile with two possible wind directions either from the west or east, which is nearly parallel to the canyon orientation. All compared model runs show a quite good agreement with the measured profile of wind speed, however, the jump profile of the wind direction is neither reproduced by the COSMO M3, nor by the CAIRDIO L0 simulation. Strikingly is, however, the nearly spot-on result of the nested CAIRDIO simulation with $5\,\mathrm{m}$ grid spacing. Note that the jumps are already seen in the nested run with $20\,\mathrm{m}$ grid spacing (not shown). Obviously, the $40\,\mathrm{m}$ run averages the wind field over a large area that is not yet representative to the specific location inside the street canyon, but nevertheless this simulation provided accurate boundary conditions for the nested simulation. Lastly, in Fig. B1e-f also measured air temperature is compared with respective profiles from the models. Notably is that both measured profiles are quite similar, despite the significantly different environments the stations are located in. Generally speaking for both sites, the modeled profiles follow the measured trend quite well, even though some specific deviations are noticeable. Firstly, during the morning of the first day, the models show

a delayed climb in temperature, and also underestimate air temperature by up to $2\,\mathrm{K}$ after midday of March 1 throughout the rest of the day until early March 2. Secondly, a sudden short-term drop in temperature of $3\,\mathrm{K}$ is measured around 1 March 12:00 UTC, but is absent in the simulations. A comparison of modeled rain rates with observation (Fig. 10) reveals that $1.2\,\mathrm{mm}$ of precipitation fell over a wider area during this specific time, while no precipitation occurred in the model at the same time. While the observed precipitation totals had a negligible impact on air quality, an impact on near-surface air temperature could

still be seen. Finally, it is still worth mentioning that the differences between the different models are small, i.e. the CAIRDIO simulations do not significantly perform better than the COSMO simulation. This indicates that while the surface-temperature downscaling described in Section 2.4.2 satisfies it's purpose to represent the thermal effects of the surface, it's accuracy still very much depends on the accuracy of the mesoscale simulation. To overcome this limitation, an own land-surface model for CAIRDIO, including a detailed parameterization of radiative interactions, would be needed.

*Author contributions.* Michael Weger contributed in the development of the model setups, the execution and evaluation of model runs, as well as in paper writing. Henriette Gebauer and Holger Baars provided the MLH estimates based on the Doppler wind lidar and Polly XT lidar data, respectively, and contributed valuable comments on the model comparison. Alfred Wiedensohler and Maik Merkel provided the BC measurements for model validation. Bernd Heinold assisted the entire modeling work and evaluation process, contributed comments to the paper's structure and content, and proofread the paper.

*Competing interests.* All authors declare that they have no competing interests.

*Acknowledgements.* Recent German-wide emission data were provided by the German Environment Agency (Umweltbundesamt, UBA). Line emissions for road transport in Leipzig were provided by the Saxon State Office for Environment, Agriculture and Geology ( Sächsisches Landesamt für Umwelt, Landwirtschaft und Geologie, LfULG), Environment and Transport Information System (Fachinformationssystem Umwelt und Verkehr, FIS UUV). In this regard, we also thank Mario Anhalt from the Office of Environmental Protection of the City of

910 Leipzig for his valuable support. Measurements of PM10 and other meteorological variables from the consulted air-monitoring sites were provided by the LfULG. Building geometries and orography (DGM1) are available from the State Enterprise for Geographic Information and Surveying Saxony (GeoSN). We thank Johannes Bühl for the Doppler-lidar based horizontal wind profiles used in the model validation. The Doppler wind lidar was funded by BMBF under FKZ: 01LKL1603A. The precipitation measurements and radiosonde data used in the paper were downloaded from the CDC-OpenData platform of the Deutscher Wetterdienst (DWD). We thank the DWD for good cooperation

and support.

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

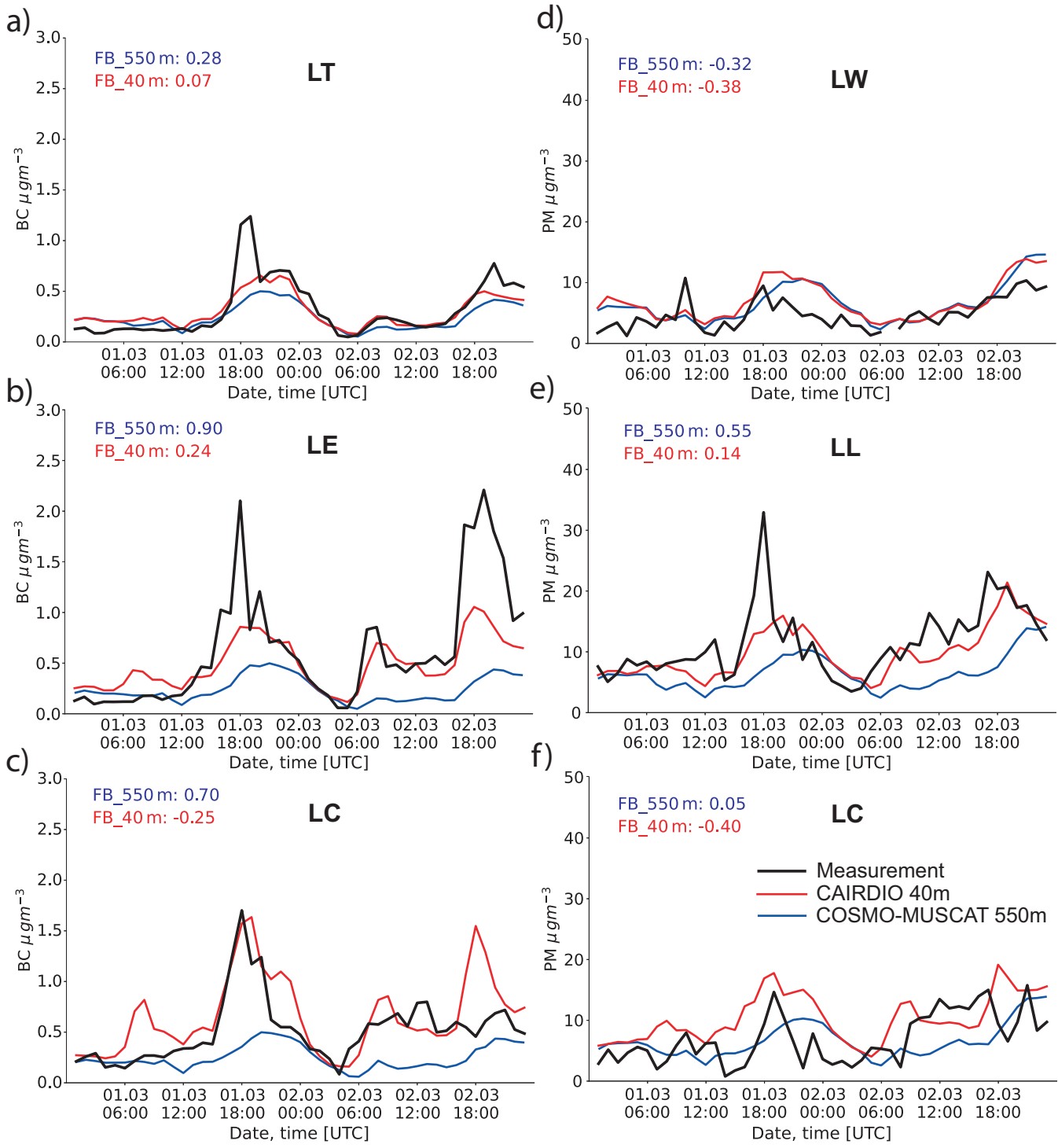

**Figure 15.** Comparison of measured BC concentrations (black lines) at the sites LE (a), LC (b), and LT (c) with simulation results of CAIRDIO L0 (red lines) and COSMO-MUSCAT M3 (blue lines). Respective profiles for $PM_{10}$ are shown for the sites LL (d), LC (e), and LW (f). Additionally shown are temporally averaged fractional bias (FB) values in relation to measurements for all model runs.

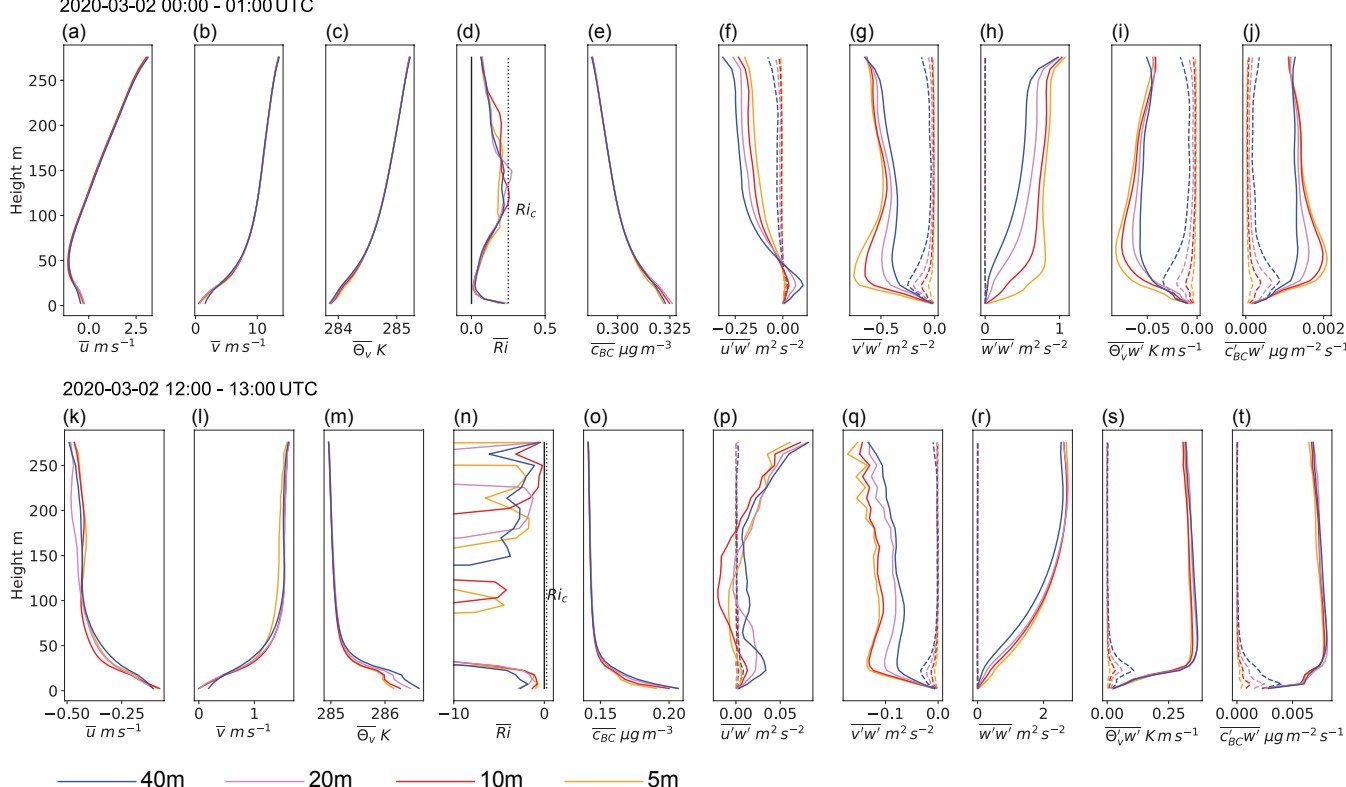

**Figure 16.** Vertical plots of horizontally averaged statistics for two different dates: Velocity components u (a, k) and v (b, l), virtual potential temperature (c, m), Richardson number (d, n), concentration of BC (e, o), and the turbulent statistics for vertical mixing of momentum (f, g, h, p, q, r), virtual potential temperature (i, s) and BC (j, t), respectively. For the turbulent statistics, dashed lines are for the subgrid-scale (sgs) contribution, while the solid lines show both the sgs and grid scale contributions.

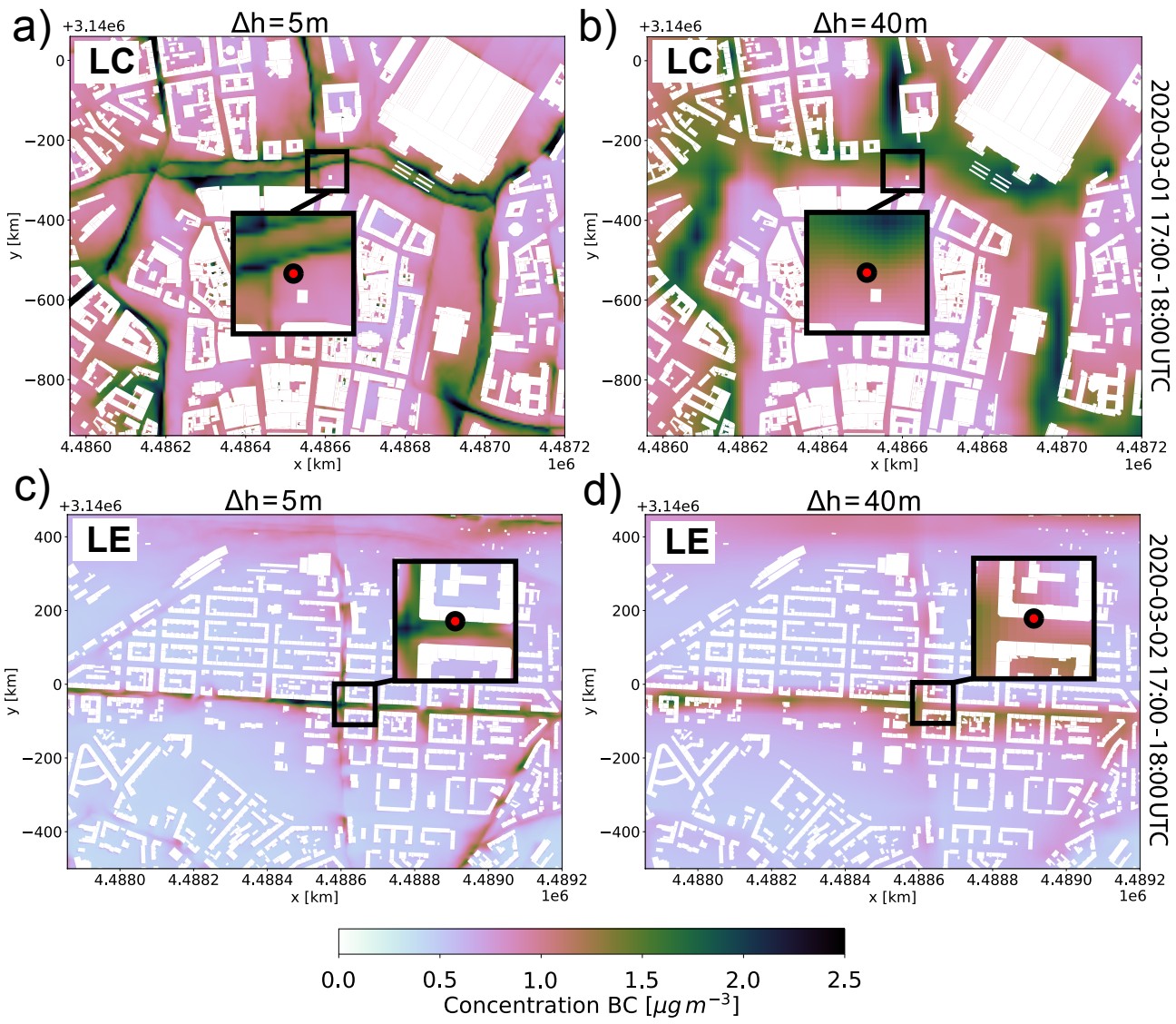

**Figure 17.** Qualitative comparison of the spatial distribution of modeled BC between the finest tested horizontal grid spacing of $5\,m$ (a) and the default $40\,m$ (b) with domain LC. Model results are temporally averaged for the rush-hour period on the evening of 1 March, and spatially interpolated at the height of corresponding air-monitoring site. In the locally magnified view of the insets, the air monitoring site is marked by a black circle. In (c) and (d) a similar comparison is shown with domain LE featuring the second rush-hour peak in the evening of 2 March.

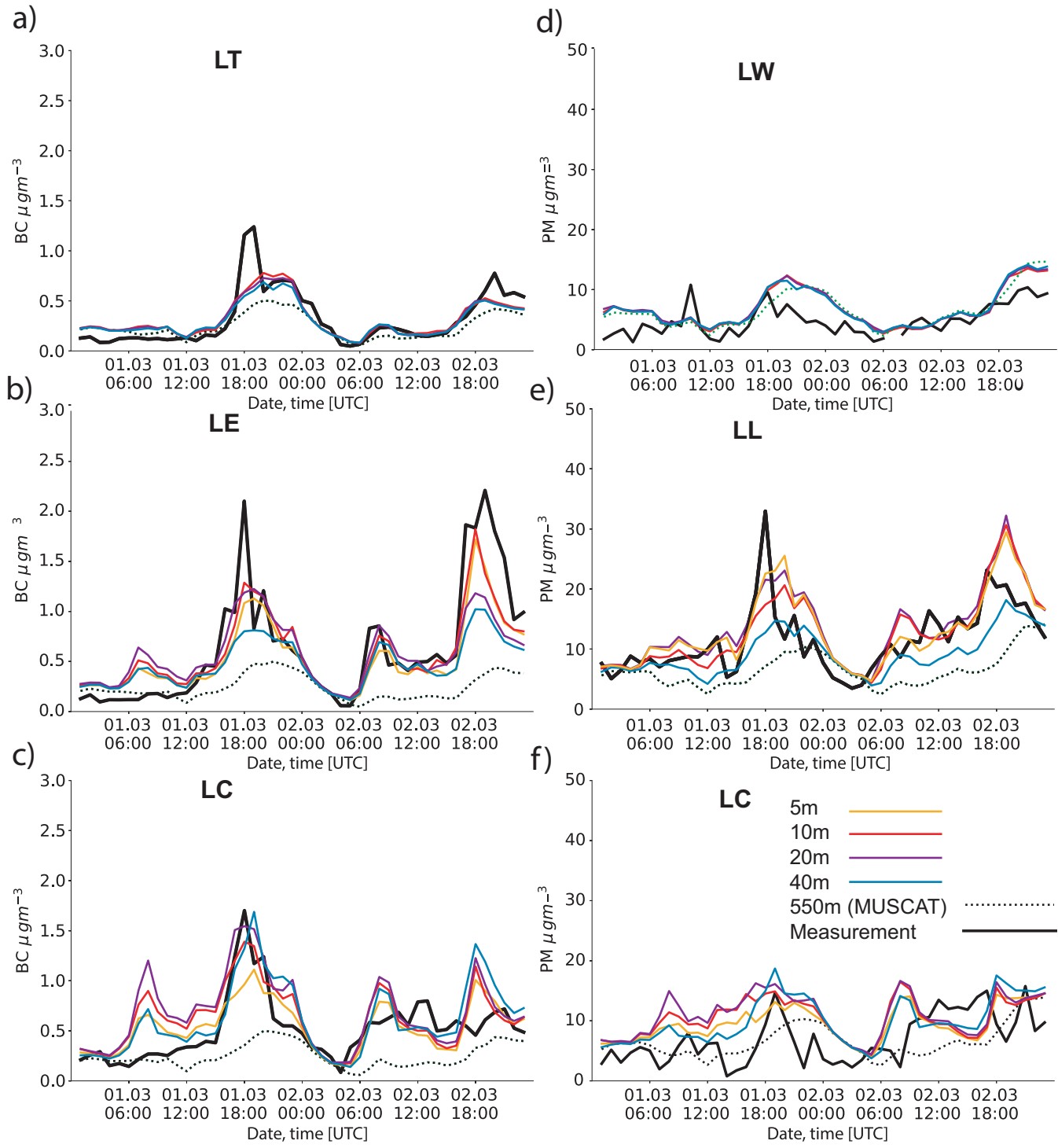

**Figure 18.** As Fig. 15, but showing the results of model sensitivity to the grid spacing. Compared are the nested CAIRDIO simulations with horizontal grid spacings of $40\,\mathrm{m}$, $20\,\mathrm{m}$, $10\,\mathrm{m}$, and $5\,\mathrm{m}$, respectively, as well as the COSMO-MUSCAT simulation with $550\,\mathrm{m}$ horizontal resolution.

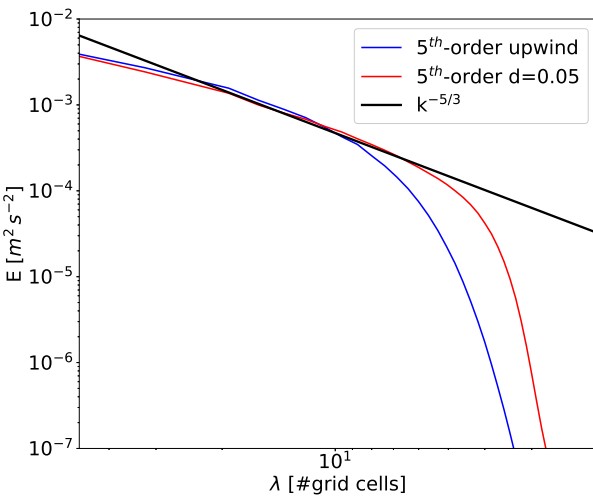

**Figure A1.** Developed energy-spectra in the flow-parallel direction for simulations of decaying isotropic turbulence with a superimposed translation velocity of similar magnitude as the turbulent fluctuations. The energy spectra are influenced by both the diffusive and dispersive errors of the advection scheme.

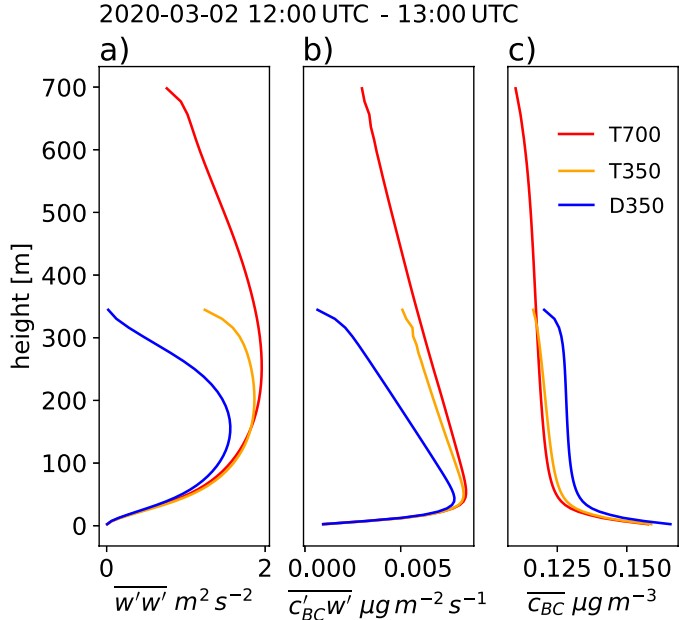

**Figure A2.** Sensitivity of the vertical turbulent (resolved plus unresolved) fluxes (a) $\overline{w'w'}$ and (b) $\overline{cBC'w'}$ on the domain height and the domain-top boundary condition, as well as impact on (c) the mean concentration profile of BC. Run T350 uses the turbulent boundary condition described in the main text of Section A0.3 in combination with the default domain height of 350 m. Run D350 also uses the default domain height, but a Rayleigh damping layer at the domain top. Run T700 uses same model settings as run T350, but the domain height is doubled to 700 m.

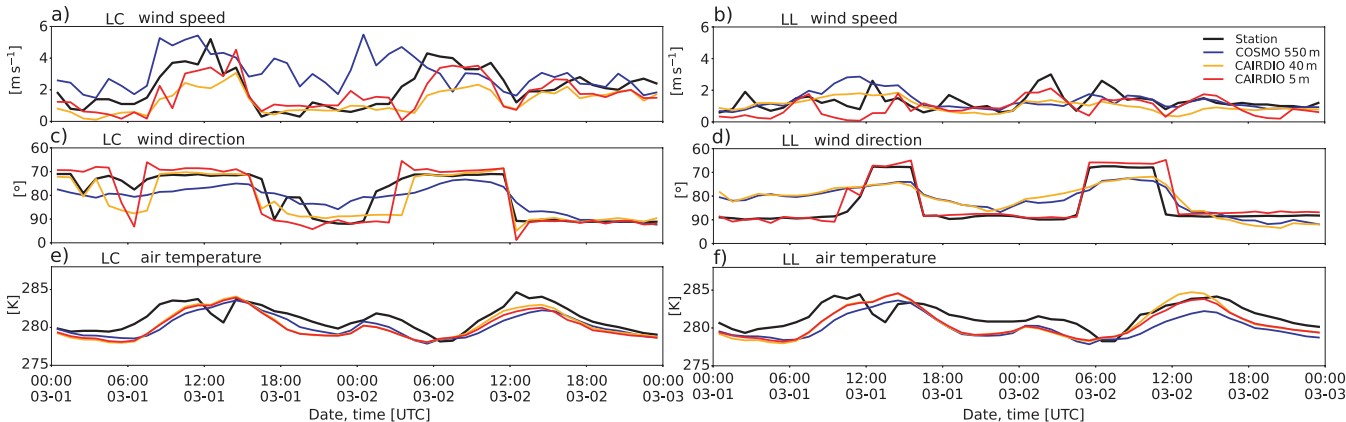

**Figure B1.** Model comparison of hourly-averaged near-surface observations (black lines) wind speed, wind direction, and air temperature with observations at the air monitoring sites LL (a, c, e) and LC (b, d, f). In the comparison included are the simulations COSMO M3 (blue lines), CAIRDIO L0 40 m (orange lines) and CAIRDIO LL, respective LC, with 5 m grid spacing (red lines).