# Peer review of "On the application and grid-size sensitivity of the urban dispersion model CAIRDIO v2.0 under real city weather conditions"

_Geoscientific Model Development, 2021_

## Author Comment (AC1)

**Response to comments of Anonymous Referee #1**

Lines 49-51: There are also studies on the combined effects of buildings and vegetation, e.g., https://doi.org/10.1016/j.buildenv.2018.09.014 and https://doi.org/10.1016/j.ufug.2016.03.006. Please add some introduction on the impacts of street vegetation on dispersion here.

It is true that urban vegetation has an impact on the wind field too, even though a much smaller one than buildings have. We added two studies in the introduction, one which directly investigates such effects, and another one, which focuses on the sensitivity on how urban trees are represented (directly, by the roughness approach, or not at all). In our case, trees and other urban vegetation are simply represented by the surface-roughness approach, which is for sure not the most accurate one. But any more complex approach (e.g. represented by diffuse obstacles) would demand a careful evaluation, preferably using wind-tunnel data. This is beyond the scope of this study but can be addressed in a future model version.

Lines 65-67: The authors might also need to mention the potential buoyancy induced by heating between buildings, e.g., https://doi.org/10.1016/j.buildenv.2012.08.029.

The effects of buoyancy from heating of building walls and ground surfaces were described in more detail and the suggested study was gratefully added.

Line 130 and Fig. 1d-f: Some modeling details should be mentioned here.

The air-quality simulations shown at this point are described in detail in Section 2.3 'Mesoscale air-quality modeling'. What was missing at this point was a reference to Section 2.3. This is now included in the text, as well as in the caption of the figure. This avoids duplication of information.

Lines 212-218: Please elaborate on the emission datasets used in model simulations. What are the spatial and temporal resolutions of these datasets? How did the authors reconcile the resolution mismatch?

This was in fact not so well elaborated. The requested information was added in Section 2.3 for the COSMO-MUSCAT model and Section 2.4.1 for the CAIRDIO L0 simulation. The resolution mismatch between the original dataset of area pollution sources (500m x 500m) and the CAIRDIO simulation (40m) was reconciled by a special downscaling method using the building geometries, as described in Section 2.4.1. Moreover, for traffic and railway emission there is no such mismatch because of the representation as line-sources. Remaining area emissions (e.g. from farming and other mobile activities) are simply interpolated conservatively to the corresponding cell area..

Lines 225-250: The relationship between M domains and L domains is unclear. Although further details can be found in the authors' previous work, a concise summary of the model is still necessary. For example, how are the buildings "effectively represented as diffuse obstacles"?

The relationship between M domains and L domains follows clearly from the different models applied (M: mesoscale, L: local). M domains are simulated with the mesoscale model COSMO-MUSCAT. The final M domain (M3) delivers the boundary conditions for the first L domain (L0) simulated with the CAIRDIO model. The coupling is described in detail in Section 2.4.2. To respond to the Reviewer's request we added a concise summary of CAIRDIO at the beginning of Section 2.4.1, which also includes a description of "diffuse obstacles boundaries (DOB)". For more details about DOB, we refer to the companion paper Weger et al., (2021).

Line 266: I noticed that the authors used different interpolation methods for these 3D variables. Is this based on some sensitivity analysis?

It would make little difference using linear interpolation for all variables. Cubic interpolation preserves the spatial details a bit better than linear interpolation, but may overemphasize larger gradients. Thus it is not suitable for air pollution fields, which should remain positive-valued after interpolation. This explanation is now included.

Equation 2: This equation is unclear. Did the authors use this simple fitting method as a substitute for a land surface model? In addition, what are the forcings of the mesoscale model (domains Ms)?

This equation is in fact a linear regression model for the surface variables. It includes the different land-cover fractions as predictor variables. Yes, the results of this fitting method are used as a substitute for the land surface model. We definitely see this as a point of improvement, but we are convinced that this approach already meets the requirements of the study reasonably well. Now, we elaborate on the limitations of the approach a bit more, and explain the method, including the equation, better. We hope that it is now more easily understood.

To the second question, the mesoscale domains are simulated with the meteorological model COSMO (a reference to the model description was still missing in 2.3.) This is a full-fledged meteorological model with an own surface model, which is still in operational use in many countries.  The coarsest domains (M0, and M1) are driven with different re-analysis data from the German meteorological office Deutscher Wetterdienst (DWD). This is all explained in Section 2.3.

Lines 354-379: It is unclear how well the mesoscale model (and the CAIRDIO model) performs during the selected period. I suggest the authors add some model evaluation results (probably in supplementary).

Although we trusted the re-analysis data, we agree that this was still a big missing part. We additionally evaluated the mesoscale simulations with a rich set of remote-sensing data (observations of horizontal wind, mixed-layer height), radiosonde and lidar data (thermal stratification), and ground-based meteorological data (for the evaluation of the urban wind field).  These additional observational data are now included in a subsection of Section 2.2. Part of the model evaluation referred to the planetary boundary layer structure is now provided in Section 3.1 (and also in Figures 9 - 10). To evaluate the urban wind field, we added a supplementary section.

Lines 406-407: What are the data sources of building geometries (and land use)? Is this "30 m" here the building height averaged across the domain?

This is only roughly the average building height plus some additional margin (accounting for the roof vortex). Basically it included the layer that is most directly influenced by buildings. Note that buildings are described by a detailed building-geometry dataset with individual roof heights. The data source of the building geometries is GeoSN and for the land use data it is Pflugmacher (2018) and Banzhaf (2018). Detailed references and access information are found in Section 2.4.1 and also in the references or acknowledgments, depending on the source.

Lines 449-450: The authors attributed the underestimated spikes to observational noises. Could this be due to uncertainties in the forcing data/emission data?

This was not very clearly formulated in the text. Correct, the observed noise indicates that there are more complex sources not represented in our emissions.

Figure 13: The authors compared the results of the LES-based dispersion model and mesoscale model in this figure. If my understanding is correct, the CAIRDIO case here is the nested LES, instead of the LES with domain L0. I am curious about the performance of the LES model at L0 level, because this will be critical to demonstrating the necessity of a finer-scale dispersion model. Did the authors check the result of the online coupled LES model?

This is in fact the LES simulation on domain L0, which was compared in this figure. We added the domain labels in the caption as it was probably not clear without them. All the nested simulations (which are only offline-nested into domain L0, see Section. 2.5) are only used for the grid-spacing sensitivity study under Section 3.3.

Lines 527-530: This comparison is interesting. Is there any possible explanation in terms of the scale-dependence here?

It is not entirely clear to us what exactly your comment refers to. However, we assume that your comment refers to the slightly higher average concentrations in the 10- and 20-m runs compared to the 5- and 40-m runs. This indeed has to do with different scales. Vertical mixing in the 40m run is predominantly at the subgrid scale, while it is mostly gridscale for the 5-m run. It seems like these two runs compare very well, given the different numerical representation of mixing. However more issues are with the 10-m and 20-m runs, where subgrid-scale and grid-scale mixing is blended together. Here, it seems that the mixing length could be increased further, more towards 40 m. This was discussed in the paper, but a few lines after line 530. We changed the order in the text, so that this discussion is directly connected to the observation.

**Minor comments:**

Abstract: Please shorten the abstract to make it concise.

Abstract is now few lines shorter as many formulations could be written in a more compact way.

Table 1: "station Central" is not in this table.

It is, but under the label LC. "Station central" should be "Leipzig Center, LC". This was corrected.

Line 319: QvS is undefined. Not sure if this is a typo (QvS).
This were in fact some typos. Should always be a capital "V".

Line 604: Please explain "diffuse buildings".

The term "diffuse buildings" is not very precise in the given context.
We replaced it with "buildings represented as physical obstacles".

---

## Author Comment (AC2)

**Response to comments of Anonymous Referee #2**

1. Considering the sensitivity of BC and PM concentrations to the near-surface meteorological conditions (i.e. 10-m wind speed, 2-m temperature), I would expect to see a short evaluation of the performance of the CAIRDIO and COSMO models for these variables (can be added as supplementary material). Such an evaluation will allow the reader to have confidence in the ability of the CAIRDIO and COSMO models to correctly predict the near-surface atmospheric forcing and consequently the BC and PM concentrations. Moreover, it could also enhance discussion/explanation of model results for the selected stations.

As requested, a model evaluation for the meteorological part was added. We evaluated the mesoscale simulation with a rich set of observations, which consist of remote-sensing data for horizontal wind, stratification and PBL height (referring to the 4th comment). To respond more specifically to the request, we evaluated near-surface wind speed, wind direction and air temperature at two urban air monitoring sites (for which the data was available). The result of the wind-field evaluation was in line with the evaluation of air pollution concentrations in respect to that the downscaled simulation with CAIRDIO is in better agreement with observations than the much coarser COSMO simulation. The limitations with the 40m grid spacing inside narrow street canyons are also again evident. Still, for a more meaningful evaluation, we think much more wind observations are needed (both a higher observational density in space and time), which are unfortunately not available. Still, we think that the mesoscale forcing applied was quite accurate. Referring to the air temperature evaluation, we observed only marginal improvements (if any at all) with the high-resolution CAIRDIO simulation, which is likely due to the diagnostic downscaling method applied for the surface variables. This finding also supports the development an own surface model for CAIRDIO in the future (apart from your remarks in the 3rd comment).

2. It is still not clear how the hybrid boundary approach at the model top for domain L0 treats the transport of TKE and the other variables. Is there inflow from the model top based on the forcing from the larger-scale COSMO model (i.e. vertical transport of Theta or TKE)? It would be beneficial to further elaborate on this, either in the methodology section or in the appendix. On a same topic have the authors tested the impact of different heights for the domain model top on the model performance?

As suggested, we dedicated now an own Section in the appendix to this hybrid boundary approach, in order to describe it in more detail and to include also a small sensitivity study with the domain height and boundary conditions changed. Basically the approach works like a turbulence recycling scheme, in that existing turbulence near the boundary is "copied" and "pasted" on the mesoscale wind field. All other variables like TKE or Theta (which are prescribed at the domain top from the COSMO model by a Dirichlet conditions) are then actively transported across the boundary by the turbulence. While the approach can still be improved (e.g. more accurate scaling of the turbulent intensities), the sensitivity experiment clearly shows its feasibility for vertically limited domains.

3. How accurate is the use of surface temperatures from the meso-scale model for the L0 domain? Although the surface temperatures from Figure 7 seems to be okay, from a qualitative perspective, they might not be accurate as surface temperatures very much depend on the atmospheric stability and the surface energy balance. In the CAIRDIO model

atmospheric stability and turbulence will be different than in the meso-scale model, and thus surface temperature and moisture may be different. This can adversely impact the near-surface turbulent transport and thus the simulated concentrations of BC and PM. I think it would be appropriate to include a discussion on this limitation, and in the future include a land-surface parametrization to compute the surface energy balance, temperature and moisture in the CAIRDIO dispersion model rather than downscaling the surface fields from the meso-scale model.

We think the first question can now be answered with the evaluation of near-surface air temperature. We did not expect an increase of accuracy by this simple downscaling approach, which we considered as a working solution to include buoyancy effects from heated surfaces. At least on a larger scale, this seems to work quite satisfactorily. However, we agree that at the LES scale, this can be considered as quite inaccurate, as it neglects the feedback from the atmosphere on the surface. Also partial shading inside street canyons cannot be represented by this approach, which is probably why the CAIRDIO air-temperature is not more accurate than the COSMO temperature at the station sites. We elaborated on these limitations more in the paper in Section 2.4.2.

4. In Figure 9d, the boundary-layer height seems rather low during the second day of the case study period, considering also the value of the Richardson number (< -1). Have the authors evaluated the performance of the COSMO model in simulating the boundary-layer height for this period. Such an evaluation can be very beneficial as it can help with the discussion of the simulated BC and PM values.

It is true that the boundary layer was in fact much higher (~1km) on the second day as it can now be concluded from the lidar observations. We think the gradient-Richardson method, which we previously used to derive the PBL height, gave systematically too low values. After consulting some more literature on this topic, we concluded that the Parcel method is the preferable choice under convective conditions and it in fact gave quite similar PBL heights as those observed. For the non-convective periods the PBL height is now determined with the bulk-Richardson method, which computes a bulk-Richardson number for the entire boundary layer, by using differences between the surface and PBL top. This method seems more suitable for vertically inhomogeneous PBL (e.g. with intermittent stable layers) than the gradient approach, and also results in a better agreement with the observations. Nevertheless, the unanimous tenor in literature is that none of the methods mentioned is very accurate.

5. How does the CAIRDIO model ensures that turbulence in generated in the domain during the model initialization? Does the model perturbs the input fields (i.e., wind speed) to faster generate turbulence in the domain or is this done using the subgrid-scale TKE from the COSMO model.

Turbulence is initialized by disturbing potential temperature using the cell-perturbation method described in Muñoz-Esparza et al. (2015). This information was added in Section 2.4.2.

**Minor comments:**

It might be better to plot the vertical y-axis without the log-scale in figure 14 (and in some other figures), as the log-scale can mask height differences, making it difficult for the reader to distinguish differences in the profiles.

As suggested, we changed the logarithmic y-axis to a linear scale. We are, however, not sure if this is better. The intention with the log scale was to emphasize more the differences within the lower part of the boundary layer and to see the logarithmic height dependency of some variables like wind speed or concentration. But we also understand the demur.

How does the CAIRDIO model treat the intra-urban vegetation? Are trees represented as diffused obstacles as well?

Urban trees are represented by the surface-roughness approach, which uses the available land-use data in 5m resolution. This piece of information was added under Secion 2.4.1. The surface-roughness approach for urban trees is for sure not the most accurate one. Urban trees will likely be improved in a future model version.

It would be nice to include the root-mean-squared error (RMSE) in Table 3.

This error metric is now also included in Table 3. It did, however, not lead to a different conclusion.

There are some errors in Figure 14 labels (u wind labels are (a,i) in the figure but are referenced in the caption (a,j)). Please adjust.

This was adjusted.

In Figure 14 it would be nice to include the vertical profile of the heat flux as well.

The vertical heat flux is now included in the Figure and discussion.

It might be preferable to change the word "grid resolution" to "grid spacing" in the manuscript as the actual horizontal resolution is larger than the grid cell size.

We completely agree therein and changed the terminology when it refers to the nominal grid spacing.

---

## Author Response (AR2)

**Author's response to Editor's comments**

Please add the version information of CAIRDIO (v 1.0) in a title.

The version number (v 2.0) is now included in the title.

CAIRDIO deals with atmospheric dispersion and I wonder how this model deals with rapid multi-phase chemical reactions in street canyons and its implications on the simulation results.

CAIRDIO is a physical dispersion model without air chemistry. For this reason, chemistry and in particular multi-phase chemistry inside street canyons could not be tested with the model. We stress, however, that for chemically inert species or comparatively slow reactions, like the formation of secondary PM, dispersion without air chemistry is a good approximation at the street-level scale. Also please note that the model MUSCAT, which was used for the precursor simulations at the mesoscale, includes such complex air chemistry.

---

## Author Response (AR3)

**Author's response to Editor's comments**

Thank you for your revision. Can you consider that my previous comments and your replay on the multi-phase chemical reaction are addressed somewhere in your manuscript?

We apologize for not having directly responded to your comment in our paper in the first place. The requested information is now contained under Section 2.4.1 in the brief introduction of the CAIRDIO model. We hope to have responded satisfactorily to your request about multi-phase chemistry in the simulations.